# Common activation mechanism of class A GPCRs

Qingtong Zhou[1†], Dehua Yang[2,3,4†], Meng Wu[1,3,5], Yu Guo[1,3,5], Wanjing Guo[2,3,4], Li Zhong[2,3,4], Xiaoqing Cai[2,4], Antao Dai[2,4], Wonjo Jang[6], Eugene I Shakhnovich[7], Zhi-Jie Liu[1,5], Raymond C Stevens[1,5], Nevin A Lambert[6], M Madan Babu[8]*, Ming-Wei Wang[2,3,4,5,9]*, Suwen Zhao[1,5]*

[1]iHuman Institute, ShanghaiTech University, Shanghai, China; [2]The CAS Key Laboratory of Receptor Research, Shanghai Institute of Materia Medica, Chinese Academy of Sciences, Shanghai, China; [3]University of Chinese Academy of Sciences, Beijing, China; [4]The National Center for Drug Screening, Shanghai Institute of Materia Medica, Chinese Academy of Sciences, Shanghai, China; [5]School of Life Science and Technology, ShanghaiTech University, Shanghai, China; [6]Department of Pharmacology and Toxicology, Medical College of Georgia, Augusta University, Augusta, United States; [7]Department of Chemistry and Chemical Biology, Harvard University, Cambridge, United States; [8]MRC Laboratory of Molecular Biology, Cambridge, United Kingdom; [9]School of Pharmacy, Fudan University, Shanghai, China

*For correspondence:
madanm@mrc-lmb.cam.ac.uk (MMB);
mwwang@simm.ac.cn (M-WW);
zhaosw@shanghaitech.edu.cn (SZ)

†These authors contributed equally to this work

Competing interests: The authors declare that no competing interests exist.

**Abstract** Class A G-protein-coupled receptors (GPCRs) influence virtually every aspect of human physiology. Understanding receptor activation mechanism is critical for discovering novel therapeutics since about one-third of all marketed drugs target members of this family. GPCR activation is an allosteric process that couples agonist binding to G-protein recruitment, with the hallmark outward movement of transmembrane helix 6 (TM6). However, what leads to TM6 movement and the key residue level changes of this movement remain less well understood. Here, we report a framework to quantify conformational changes. By analyzing the conformational changes in 234 structures from 45 class A GPCRs, we discovered a common GPCR activation pathway comprising of 34 residue pairs and 35 residues. The pathway unifies previous findings into a common activation mechanism and strings together the scattered key motifs such as CWxP, DRY, Na$^+$ pocket, NPxxY and PIF, thereby directly linking the bottom of ligand-binding pocket with G-protein coupling region. Site-directed mutagenesis experiments support this proposition and reveal that rational mutations of residues in this pathway can be used to obtain receptors that are constitutively active or inactive. The common activation pathway provides the mechanistic interpretation of constitutively activating, inactivating and disease mutations. As a module responsible for activation, the common pathway allows for decoupling of the evolution of the ligand binding site and G-protein-binding region. Such an architecture might have facilitated GPCRs to emerge as a highly successful family of proteins for signal transduction in nature.

## Introduction

As the largest and most diverse group of membrane receptors in eukaryotes, GPCRs mediate a wide variety of physiological functions (*Lagerström and Schiöth, 2008*; *Rosenbaum et al., 2009*; *Katritch et al., 2012*; *Venkatakrishnan et al., 2013*; *Katritch et al., 2013*), including vision, olfaction, taste, neurotransmission, endocrine and immune responses via more than 800 family members, and are involved in many diseases (*Rana et al., 2001*; *Smit et al., 2007*; *Vassart and Costagliola,*

*2011*; *Thompson et al., 2014*; *Hauser et al., 2018*). Therefore, GPCRs are important drug targets. There are 475 marketed drugs (~34% of all FDA-approved therapeutic agent agents) targeting 108 members of the GPCR superfamily (*Hauser et al., 2018*; *Hauser et al., 2017*; *Allen and Roth, 2011*). Class A is the largest and most diverse GPCR subfamily in humans (*Kolakowski, 1994*; *Bockaert and Pin, 1999*; *Fredriksson et al., 2003*; *Isberg et al., 2016*), including 388 olfactory (*Krautwurst et al., 1998*; *Spehr and Munger, 2009*) and 286 non-olfactory receptors (*Pándy-Szekeres et al., 2018*; *Munk et al., 2019*) (*Figure 1a*). They share a seven-transmembrane (7TM) helices domain, with ligand binding pocket and G-protein-binding region located in the extracellular and intracellular ends of the helix bundle. Responding to a wide variety of extracellular signals ranged from small molecules to peptides even proteins, the extracellular facing ligand-binding pockets have evolved to be highly diverse in both shape and sequences (*Venkatakrishnan et al., 2013*; *Ngo et al., 2017*; *Vass et al., 2018*). Similarly, the G-protein-binding regions are also quite diverse in sequences, modulating the activity of different signalling pathways by recruiting dozens of hetero-trimeric G proteins (*Rasmussen et al., 2011a*; *Du et al., 2019*; *Liu et al., 2019a*), arrestins (*Gainetdinov et al., 2004*; *Zhou et al., 2017*; *Yang et al., 2018*; *Latorraca et al., 2018*), GPCR kinases (*Komolov et al., 2017*) in a ligand-specific manner. Residues that connect the ligand-binding pocket to the G-protein-coupling region are significantly more conserved (*Ballesteros and Weinstein, 1995*; *Isberg et al., 2015*), with evolutionarily conserved sequence motifs (CWxP [*Eddy et al., 2018*; *Filipek, 2019*; *Wescott et al., 2016*; *Tehan et al., 2014*; *Holst et al., 2010*; *Nygaard et al., 2009*; *Hofmann et al., 2009*; *Trzaskowski et al., 2012*], PIF [*Ballesteros and Weinstein, 1995*; *Ishchenko et al., 2017*; *Schönegge et al., 2017*; *Kato et al., 2019*] Na[+] pocket [*Eddy et al., 2018*; *Filipek, 2019*; *Liu et al., 2012*; *Yuan et al., 2013*; *Fenalti et al., 2014*; *Katritch et al., 2014*;

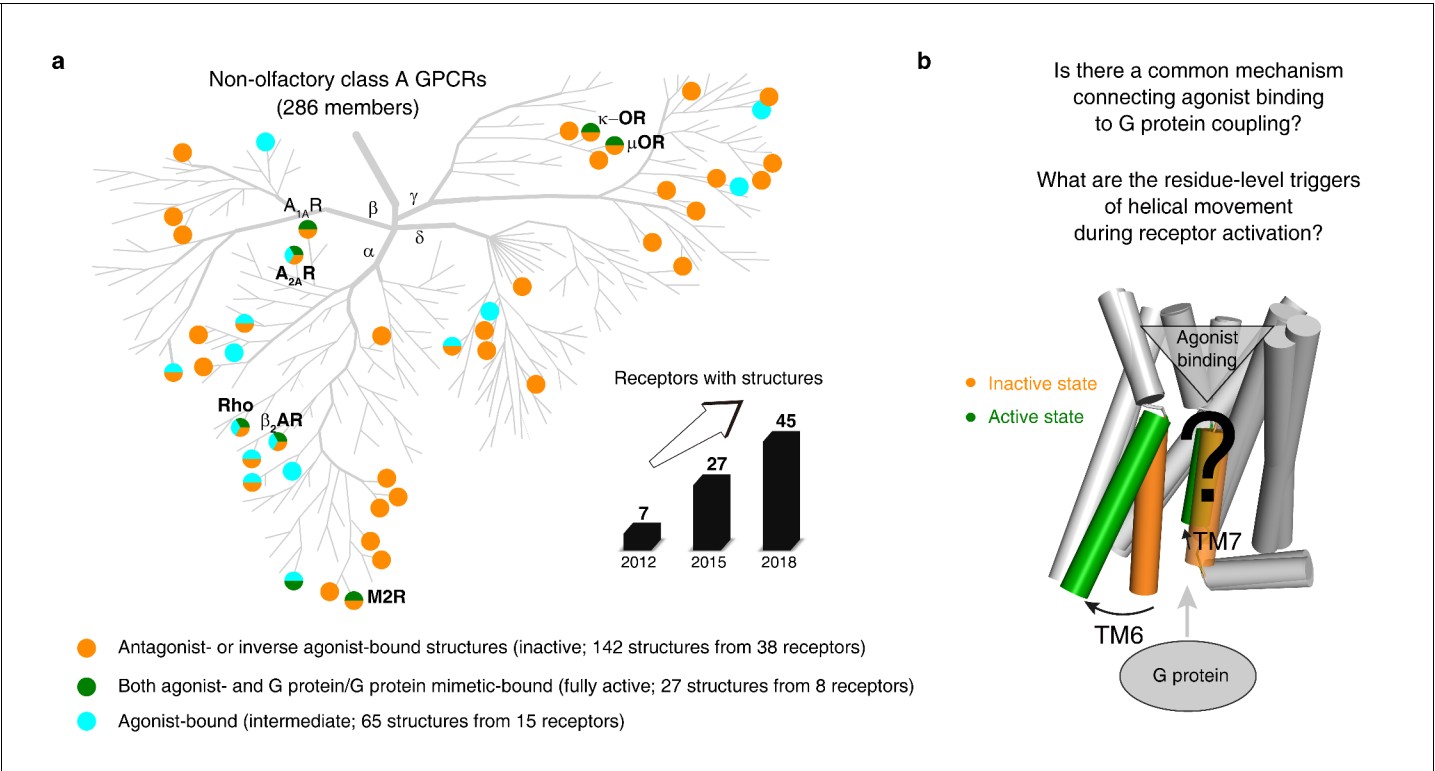

**Figure 1.** An increasing number of reported class A GPCR structures facilitates studies on common activation mechanism. (**a**) Distribution of structures in different states in the non-olfactory class A GPCR tree as of October 1, 2018. (**b**) Common GPCR activation mechanism and the residue-level triggers are not well understood.

The online version of this article includes the following source data and figure supplement(s) for figure 1:

**Source data 1.** The released class A GPCR structures (as of October 1, 2018).
**Source data 2.** Disease mutations occurred in class A GPCRs.
**Figure supplement 1.** The pattern of conservation of residues and the map of number of disease-associated mutations in human class A GPCRs.

*Vickery et al., 2018*; *White et al., 2018*; *Ye et al., 2018*; *Chen et al., 2019*], NPxxY [*Rasmussen et al., 2011a*; *Filipek, 2019*; *Wescott et al., 2016*; *Nygaard et al., 2009*; *Hofmann et al., 2009*; *Trzaskowski et al., 2012*; *Schönegge et al., 2017*; *Chen et al., 2019*; *Venkatakrishnan et al., 2016*] and DRY [*Schönegge et al., 2017*; *Alhadeff et al., 2018*; *Jacobson et al., 2014*; *Feng et al., 2017*; *Roth et al., 2017*; *Shihoya et al., 2017*; *Yuan et al., 2014*]) scattered in the intracellular half of the 7TM domain (*Figure 1—figure supplement 1*).

GPCR activation is agonist binding induced G-protein recruitment (*Eddy et al., 2018*; *Christopoulos and Kenakin, 2002*; *Manglik et al., 2015*; *Staus et al., 2016*; *DeVree et al., 2016*). It is an allosteric process (*Gilchrist, 2007*; *May et al., 2007*; *Christopoulos, 2014*; *Latorraca et al., 2017*), transducing various external stimuli into cellular responses. Understanding the activation mechanism of GPCR is of paramount importance in pharmacology research and drug discovery. Tremendous previous efforts, involving sequence analysis, structural, biophysical, biochemical and computational approaches such as X-ray (*Popov et al., 2018*; *Roth et al., 2008*; *Warne et al., 2009*; *Carpenter et al., 2016*; *Nehmé et al., 2017*; *Tsai et al., 2018*; *Rasmussen et al., 2011b*; *Pardon et al., 2014*; *Rosenbaum et al., 2007*; *Kobilka and Schertler, 2008*; *Cherezov et al., 2007*; *Ghosh et al., 2015*; *Cherezov et al., 2004*; *Caffrey and Cherezov, 2009*; *Liu et al., 2013*; *Weierstall et al., 2014*; *Stauch and Cherezov, 2018*; *Zhang et al., 2015*), NMR (*Ye et al., 2018*; *Chen et al., 2019*; *Nygaard et al., 2013*; *Lamichhane et al., 2015*; *Isogai et al., 2016*; *Sounier et al., 2015*; *Ye et al., 2016*; *Shimada et al., 2019*), Cryo-EM (*Liang et al., 2017*; *Zhang et al., 2017*; *Cheng, 2018*; *Renaud et al., 2018*), labeling biosensors (*Irannejad et al., 2013*; *Tian et al., 2017*), FRET (*Gregorio et al., 2017*; *Halls and Canals, 2018*; *Sandhu et al., 2019*), BRET (*Lan et al., 2012*; *Lee et al., 2016*; *Okashah et al., 2019*), DEER (*Wingler et al., 2019*; *Van Eps et al., 2018*; *Dror et al., 2015*), molecular dynamic simulations (*Yuan et al., 2013*; *Dror et al., 2015*; *Dror et al., 2011a*; *Dror et al., 2011b*; *Dror et al., 2013*; *Miao et al., 2013*; *Bhattacharya and Vaidehi, 2014*; *Kohlhoff et al., 2014*; *Ciancetta et al., 2015*; *Alhadeff et al., 2018*), evolutionary tracing (*Madabushi et al., 2004*; *Schöneberg et al., 2007*; *Rodriguez et al., 2010*), molecular docking (*Kufareva et al., 2011*; *Jacobson et al., 2014*; *Kooistra et al., 2016*; *Miao et al., 2016*; *Feng et al., 2017*; *Roth et al., 2017*; *Lyu et al., 2019*; *Cooke et al., 2015*) and mutagenesis (*Schönegge et al., 2017*; *Sung et al., 2016*; *Massink et al., 2015*; *Ragnarsson et al., 2019*; *Hulme, 2013*) have been made to study the allosteric nature of GPCRs including but not limited to receptor activation (*Wescott et al., 2016*; *Tehan et al., 2014*; *Trzaskowski et al., 2012*; *Venkatakrishnan et al., 2016*; *Isom and Dohlman, 2015*; *Okada et al., 2001*; *Hunyady et al., 2003*; *Dalton et al., 2015*; *Lans et al., 2015*), G protein activation (*Trzaskowski et al., 2012*; *Flock et al., 2017*; *Flock et al., 2015*; *Furness et al., 2016*; *Glukhova et al., 2018*; *Ilyaskina et al., 2018*; *Inoue et al., 2019*; *Weis and Kobilka, 2018*), biased agonism (*McCorvy et al., 2018*; *Onaran et al., 2014*; *Schmid et al., 2017*; *Smith et al., 2018*; *Tan et al., 2018*; *Whalen et al., 2011*; *Wootten et al., 2018*; *Wootten et al., 2016*; *Masureel et al., 2018*), ligand efficiency (*Gregorio et al., 2017*; *Furness et al., 2016*; *Livingston et al., 2018*; *Solt et al., 2017*; *Yao et al., 2009*), allosteric modulators (*Thal et al., 2018*; *Kruse et al., 2013*; *Leach et al., 2007*; *Liu et al., 2019b*; *Lu and Zhang, 2019*; *Robertson et al., 2018*; *Shao et al., 2019*; *Wu et al., 2019*; *Zheng et al., 2016*; *Ortiz Zacarías et al., 2018*; *Jaeger et al., 2019*; *Hollingsworth et al., 2019*; *Liu et al., 2017*; *Chaturvedi et al., 2018*; *Oswald et al., 2016*), and inverse agonism (*Chaturvedi et al., 2018*; *Oswald et al., 2016*; *Hori et al., 2018*; *Nagiri et al., 2019*; *Peng et al., 2018*; *Shihoya et al., 2017*). Starting from the first structure of a GPCR-G protein complex (β$_2$AR-G$_s$) (*Rasmussen et al., 2011a*), the rapidly growing structures of receptor-G-protein complex have provided excellent opportunity to better understand receptor conformation changes upon activation. Meanwhile, mutagenesis studies on different receptors also identified functional roles of key residues in receptor activation, one good example is CXCR4 (*Wescott et al., 2016*), where a large-scale mutagenesis study covering all 352 residues of the receptor identified 41 amino acids that are required for signalling induced by agonist CXCL12. Notably, family-wide analysis on GPCR activation with the concept of residue contacts (*Venkatakrishnan et al., 2013*; *Flock et al., 2017*; *Kayikci et al., 2018*) have revealed the converged activation pathway near the G-protein-coupling region (*Venkatakrishnan et al., 2016*) and selectivity determinants of GPCR–G-protein binding (*Flock et al., 2017*). While these studies have provided key insights into GPCR activation mechanism for individual receptors or specific motifs, a family-wide common activation mechanism that directly connect ligand-binding pocket and G-protein-coupling region has yet to be discovered. Although it

is well established that outward movement of transmembrane helix 6 (TM6) upon ligand binding is a common feature of receptor activation, the residue level changes that trigger the movement of TM6 remain less well understood (*Figure 1b*).

Receptor activation requires global reorganization of residue contacts as well as water-mediated interactions (*Yuan et al., 2014*; *Yuan et al., 2016*; *Venkatakrishnan et al., 2019*). Since prior studies primarily investigated conformational changes through visual inspection (*Tehan et al., 2014*; *Trzaskowski et al., 2012*) or through the presence or absence of non-covalent contacts between residues (*Venkatakrishnan et al., 2016*; *Flock et al., 2017*), we reasoned that one could gain comprehensive knowledge about mechanism of receptor activation by developing approaches that can capture not just the presence or absence of a contact but also subtle, and potentially important alterations in conformations upon receptor activation.

## Results

### A residue-residue contact score-based framework to characterize GPCR conformational changes

To address this, we developed an approach to rigorously quantify residue contacts in proteins structures and infer statistically significant conformational changes. We first defined a residue-residue contact score (RRCS) which is an atomic distance-based calculation that quantifies the strength of contact between residue pairs (*Ngo et al., 2017*) by summing up all possible inter-residue heavy atom pairs (*Figure 2a* and *Figure 2—figure supplement 1a*). We then defined ΔRRCS, which is the difference in RRCS of a residue pair between any two conformational states of a receptor that quantitatively describes the rearrangements of residue contacts (*Figure 2b* and *Figure 2—figure supplement 1b*). While RRCS can be 0 (no contact) or higher (stronger contact), ΔRRCS can be negative (loss in strength of residue contact), positive (gain in strength of residue contact) or 0 (no change in strength of residue contact). To capture the entirety of conformational changes in receptor structure upon activation, we computed the ΔRRCS between the active and inactive states of a receptor and defined two types of conformational changes (*Figure 2c*): (i) switching contacts: these are contacts that are present in the inactive state but lost in the active state (or vice versa) such as loss of intra-helical contacts between D/E$^{3\times49}$ (GPCRdb numbering [*Isberg et al., 2016*]) and R$^{3\times50}$, and gain of inter-helical hydrophobic contacts between residues at $3\times40$ and $6\times48$ upon receptor activation; and (ii) repacking contacts: these are contacts that result in an increase or decrease in residue packing such as the decreased packing of intra-helical side-chain contacts between W$^{6\times48}$ and F$^{6\times44}$, and the increase in inter-helical residue packing due to the translocation of N$^{7\times49}$ toward D$^{2\times50}$ upon receptor activation. In this manner, we quantified the global, local, major and subtle conformational changes in a systematic way (i.e. inter-helical and intra-helical, switching and repacking contacts).

We then analyzed 234 structures of 45 class A GPCRs that were grouped into three categories (*Figure 1a*): (i) antagonist- or inverse agonist-bound (inactive; 142 structures from 38 receptors); (ii) both agonist- and G protein/G protein mimetic-bound (fully active; 27 structures from eight receptors); and (iii) agonist-bound (intermediate; 65 structures from 15 receptors). Among them, six receptors [rhodopsin (bRho) (*Li et al., 2004*; *Choe et al., 2011*), β$_2$-adrenergic receptor (β$_2$AR) (*Rasmussen et al., 2011a*; *Cherezov et al., 2007*), M2 muscarinic receptor (M2R) (*Kruse et al., 2013*; *Haga et al., 2012*), μ-opioid receptor (μOR) (*Manglik et al., 2012*; *Huang et al., 2015*), adenosine A$_{2A}$ receptor (A$_{2A}$R) (*Carpenter et al., 2016*; *Jaakola et al., 2008*) and κ-opioid receptor (κ-OR) (*Wu et al., 2012*; *Che et al., 2018*)] have both inactive- and active-state crystal structures available. Given that ΔRRCS can capture major and subtle conformational changes, we computed RRCS for all structures and ΔRRCS for the six pairs of receptors and investigated the existence of a common activation pathway (i.e. a common set of residue contact changes) across class A GPCRs. Two criteria (*Figure 2d*; further details in Materials and methods) were applied to identify conserved rearrangements of residue contacts: (i) equivalent residue pairs show a similar and substantial change in RRCS between the active and inactive state structures of each of the six receptors (i.e. the same sign of ΔRRCS and |ΔRRCS| > cut-off for all receptors) and (ii) family-wide comparison of the RRCS for the 142 inactive and 27 active state structures shows a statistically significant difference (p<0.001; two sample *t*-test). This allowed us to reliably capture both the major rearrangements as well as subtle but conserved conformational changes at the level of individual residues in diverse GPCRs in a

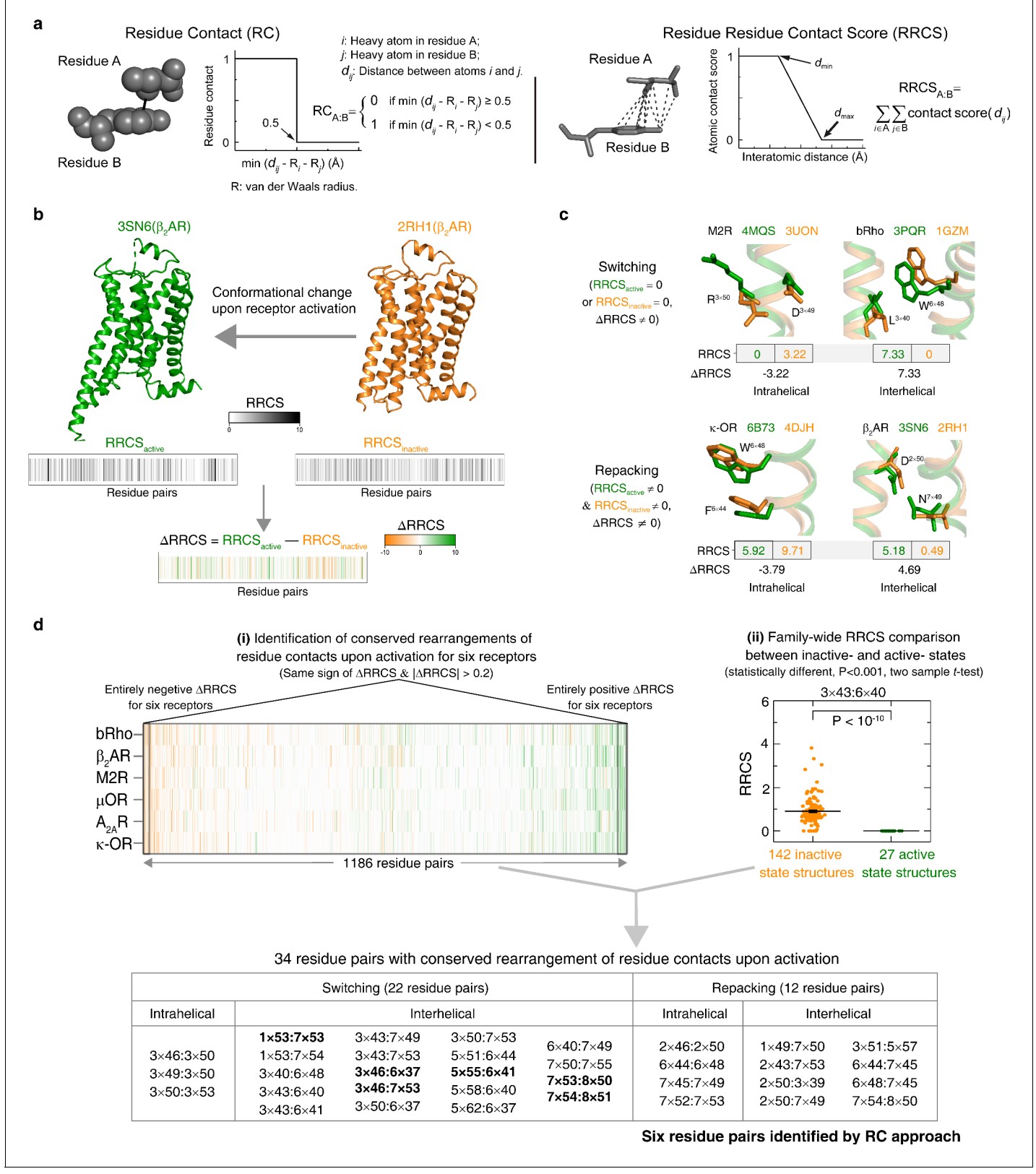

**Figure 2.** Understanding GPCR activation mechanism by RRCS and ΔRRCS. (**a**) Comparison of residue contact (RC) (***Venkatakrishnan et al., 2016***) and residue residue contact score (RRCS) calculations. RRCS can describe the strength of residue-residue contact quantitatively in a much more accurate manner than the Boolean descriptor RC. (**b**) RRCS and ΔRRCS calculation for a pair of active and inactive structures can capture receptor

*Figure 2 continued on next page*

*Figure 2 continued*

conformational change upon activation. (c) Two types of conformational changes (i.e. switching and repacking contacts) can be defined by RRCS to quantify the global, local, major and subtle conformational changes in a systematic way. (d) Two criteria of identifying conserved residue rearrangements upon receptor activation by RRCS and ΔRRCS. Thirty-four residues pairs were identified based on the criteria (see Materials and methods, *Figure 2—source datas 1* and *2* for details), only six of them were discovered before (*Venkatakrishnan et al., 2016*).

The online version of this article includes the following source data and figure supplement(s) for figure 2:

**Source data 1.** Calculated RRCS of 34 residue pairs constituting the common activation pathway for released class A GPCR structures.
**Source data 2.** Thirty-four residue pairs show conserved rearrangements of residue contacts upon activation.
**Figure supplement 1.** Calculation of RRCS and ΔRRCS.

statistically robust and significant manner. Consistent with this, a comparison with earlier studies revealed that the RRCS based approach is able to capture a larger number of conserved large-scale and subtle changes in residues contacts (*Figure 2d*) that would have been missed by visual inspection or residue contact presence/absence criteria alone (see Materials and methods for conceptual advance of this approach and detailed comparison).

## Discovery of a common and conserved receptor activation pathway

Remarkably, for the first time, our analysis of the structures allowed the discovery of a common and conserved activation pathway that directly links ligand-binding pocket and G protein-coupling regions in class A GPCRs (*Figure 3*). The pathway is comprised of 34 residue pairs (formed by 35 residues) with conserved rearrangement of residue contacts upon activation (*Figure 2d*), connecting several well-known but structurally and spatially disconnected motifs (CWxP, PIF, $Na^+$ pocket, NPxxY and DRY) all the way from the extracellular side (where the ligand binds) to the intracellular side (where the G-protein binds). Inspection of the rewired contacts as a ΔRRCS network reveals that the conserved receptor activation pathway is of modular nature and involves conformational changes in four layers. In layer 1, there is a conserved signal initiation step involving changes in residue contacts at the bottom of the ligand-binding pocket and $Na^+$ pocket. In layer 2, critical hydrophobic contacts are broken (*i.e.*, opening of the hydrophobic lock). In layer 3, microswitch residues ($6\times37$, $Y^{7\times53}$) are rewired and in layer 4, the residue $R^{3\times50}$ and G protein contacting positions are rewired, making them competent to bind to G protein on the cytosolic side (*Figure 3*). Strikingly, recently released cryo-EM structures of four receptors (5-HT$_{1B}$, rhodopsin, A$_1$R and μOR) in complex with G$_{i/o}$(*Glukhova et al., 2018*; *García-Nafría et al., 2018*; *Kang et al., 2018*; *Koehl et al., 2018*; *Draper-Joyce et al., 2018*; *Tsai et al., 2018*) also support the conservation of contacts involving these 34 residue pairs (*Figure 4*, *Figure 4—figure supplements 1* and *2*). These observations highlight the conserved and common nature of a previously undescribed activation pathway linking ligand binding to G-protein coupling, regardless of the subtypes of intracellular effectors (*i.e.*, G$_s$ (*Rasmussen et al., 2011a*; *Carpenter et al., 2016*), G$_{i/o}$(*Glukhova et al., 2018*; *García-Nafría et al., 2018*; *Kang et al., 2018*; *Koehl et al., 2018*; *Draper-Joyce et al., 2018*; *Tsai et al., 2018*), arrestin (*Zhou et al., 2017*; *Kang et al., 2015*) or G-protein mimetic nanobody/peptide (*Rasmussen et al., 2011b*; *Kruse et al., 2013*; *Choe et al., 2011*; *Huang et al., 2015*; *Che et al., 2018*), *Figure 4a*).

Collectively, these findings illustrate how a combination of intra-helical and inter-helical switching contacts as well as repacking contacts underlies the common activation mechanism of GPCRs.

## Molecular insights into key steps of the common activation pathway

Receptor activation is triggered by ligand binding and is characterised by movements of different transmembrane helices. How does ligand-induced receptor activation connect the different and highly conserved motifs, rewire residue contacts and result in the observed changes in transmembrane helices? To this end, we analyzed the common activation pathway in detail and mapped, where possible, how they influence helix packing, rotation and movement (*Figure 3*). A qualitative analysis suggests the presence of four layers of residues in the pathway linking the ligand binding residues to the G-protein-binding region.

Layer 1: We did not see a single ligand-residue contact that exhibits conserved rearrangement, which accurately reflects the diverse repertoire of ligands that bind GPCRs (*Katritch et al., 2012*; *Venkatakrishnan et al., 2013*; *Ngo et al., 2017*) (*Figure 3—figure supplement 1*). Instead, as a first

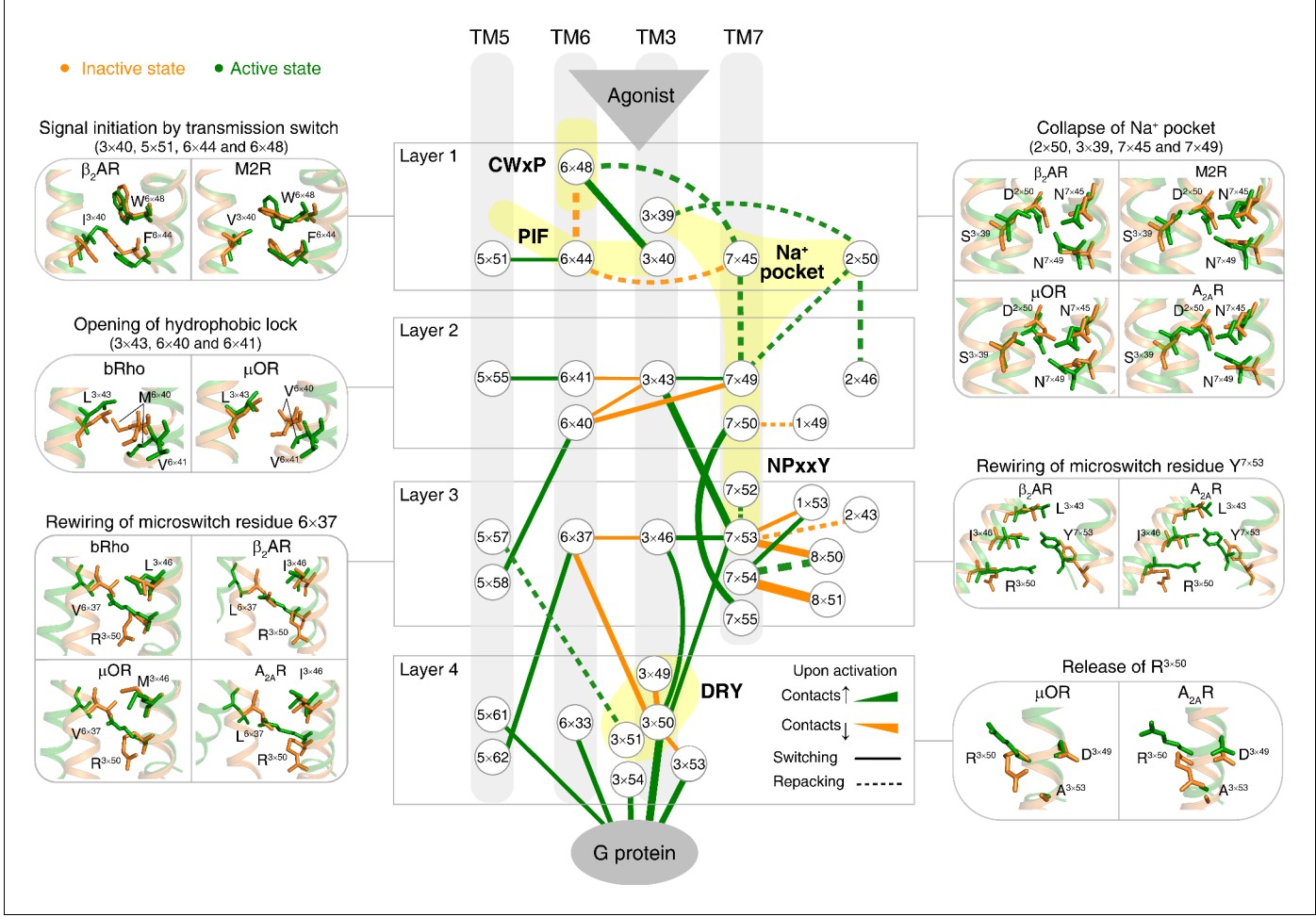

**Figure 3.** Common activation pathway of class A GPCRs. Node represents structurally equivalent residue with the GPCRdb numbering (*Isberg et al., 2016*) while the width of edge is proportional to the average ΔRRCS among six receptors (bRho, β₂AR, M2R, μOR, A₂AR and κ-OR). Four layers were qualitatively defined based on the topology of the pathway and their roles in activation: signal initiation (layer 1), signal propagation (layer 2), microswitches rewiring (layer 3) and G-protein coupling (layer 4).

The online version of this article includes the following figure supplement(s) for figure 3:

**Figure supplement 1.** Rearrangements of ligand-residue contacts in ligand-binding pocket are not conserved, reflecting diverse ligand recognition modes.

common step, extracellular binding of diverse agonists converges to trigger an identical alteration of the transmission switch ($3\times40$, $5\times51$, $6\times44$ and $6\times48$) and Na$^+$ pocket ($2\times50$, $3\times39$, $7\times45$ and $7\times49$). Specifically, the repacking of an intra-helical contact between residues at $6\times48$ and $6\times44$, together with the switching contacts of residue at $3\times40$ toward $6\times48$ and residue at $5\times51$ toward $6\times44$, contract the TM3-5-6 interface in this layer. This reorganization initializes the rotation of the cytoplasmic end of TM6. The collapse of Na$^+$ pocket leads to a denser repacking of the four residues ($2\times50$, $3\times39$, $7\times45$ and $7\times49$), initiating the movement of TM7 toward TM3. Remarkably, a recent NMR study on A₂AR (*Eddy et al., 2018*) demonstrated the strong coupling between allosteric switch D$^{2\times50}$ and toggle switch W$^{6\times48}$, which is consistent with the present observation.

Layer 2: In parallel with these movements, two residues ($6\times40$ and $6\times41$) switch their contacts with residue at $3\times43$, and form new contacts instead with residues at $5\times58$ and $5\times55$, respectively. Residues at $3\times43$, $6\times40$ and $6\times41$ are mainly composed of hydrophobic amino acids and referred as hydrophobic lock (*Wescott et al., 2016*; *Tehan et al., 2014*; *Han et al., 2012*; *Martí-Solano et al., 2014*). Its opening loosens the packing of TM3-TM6 and facilitates the outward movement of the cytoplasmic end of TM6, which is necessary for receptor activation. Additionally, N$^{7\times49}$

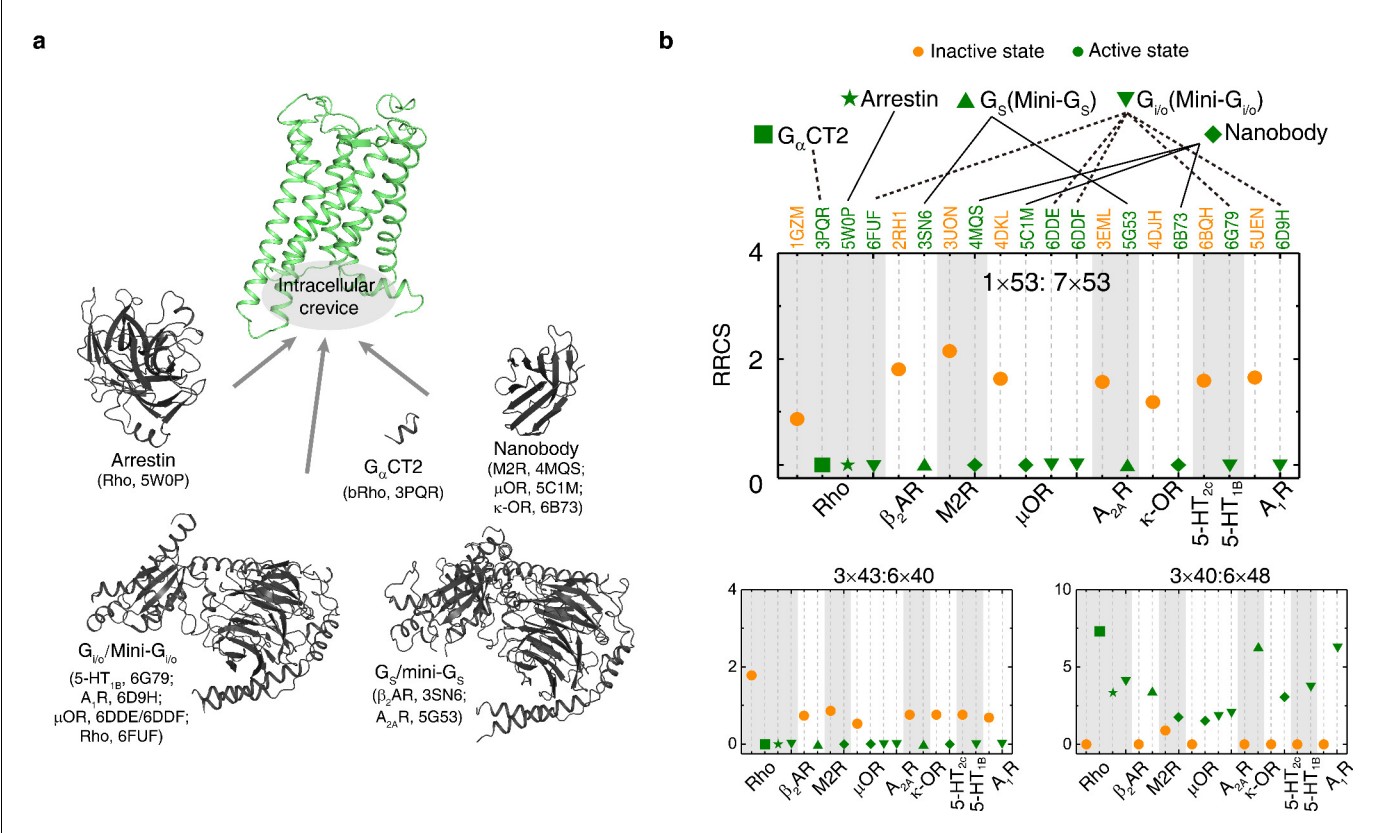

**Figure 4.** The common activation mechanism is the shared portion of various downstream pathways of different class A GPCRs. (**a**) Intracellular binding partners used in the active state structures. (**b**) Comparison of RRCS for active (green) and inactive (orange) states of eight receptors with different intracellular binding partners, including four recently solved cryo-EM structures of $G_{i/o}$-bound receptors (5-HT$_{1B}$ receptor, rhodopsin, A$_1$R and μOR) (*Tsai et al., 2018*; *García-Nafría et al., 2018*; *Kang et al., 2018*; *Koehl et al., 2018*; *Draper-Joyce et al., 2018*) whose resolutions were low (usually ≥3.8 Å for the GPCR part). Nevertheless, almost all conserved residue rearrangements in the pathway can be observed from them. Three of 34 residues pairs were shown here, see *Figure 4—figure supplements 1* and *2* for the remaining 31 residue pairs.
The online version of this article includes the following figure supplement(s) for figure 4:

**Figure supplement 1.** The switching conformation change is conserved upon receptor activation.
**Figure supplement 2.** The repacking conformation change is conserved upon receptor activation.

develops contacts with residue at 3×43 from nothing, facilitating the movement of TM7 toward TM3.

Layer 3: Upon receptor activation, $Y^{7×53}$ loses its inter-helical contacts (*Venkatakrishnan et al., 2016*) with residues at 1×53 and 8×50, and forms new contacts with residues at 3×43, 3×46 and $R^{3×50}$, which were closely packed with residues in TM6. Thus, the switching of contacts by $Y^{7×53}$ strengthens the packing of TM3-TM7, while the packing of TM3-TM6 is further loosened with the outward movement of TM6.

Layer 4: Finally, the restrains on $R^{3×50}$, including more conserved, local intra-helical contacts with $D(E)^{3×49}$ and less conserved ionic lock with $D(E)^{6×30}$, are eliminated and $R^{3×50}$ is released. Notably, the switching contacts between $R^{3×50}$ and residue at 6×37 are essential for the release of $R^{3×50}$, which breaks the remaining contacts between TM3 and TM6 in the cytoplasmic end and drives the outward movement of TM6. The rewired contacts of $R^{3×50}$ and other G-protein contacting positions (3×53, 3×54, 5×61 and 6×33) make the receptor competent to bind to G protein on the cytosolic side.

Together, these findings demonstrate that the intra-helical/inter-helical and switching/repacking contacts between residues is not only critical to reveal the continuous and modular nature of the activation pathway, but also to link residue-level changes to transmembrane helix-level changes in the receptor.

## Common activation pathway induced changes in TM helix packing

To capture the patterns in the global movements of transmembrane helices, all inter-helical residues pairs in the common activation pathway were used to describe the inter-helical contacts between the cytoplasmic end of TM3 and TM6 as well as TM3 and TM7 (*Figure 5a*). Analysis of the RRCS$_{TM3-TM7}$ (X-axis) and RRCS$_{TM3-TM6}$ (Y-axis) for each of the 234 class A GPCR structures revealed distinct compact clusters of inactive and active states. Surprisingly, the inactive state has zero or close to zero RRCS$_{TM3-TM7}$ regardless of the wide distribution of RRCS$_{TM3-TM6}$. In contrast, the active state has a high RRCS$_{TM3-TM7}$ and strictly zero RRCS$_{TM3-TM6}$. Thus, receptor activation from inactive to active state occurs as a harmonious process of inter-helical contact changes: elimination of TM3-TM6 contacts, formation of TM3-TM7 contacts and repacking of TM5-TM6 (*Figure 5b* and *Figure 5—figure supplement 1*). In terms of global conformational changes, the binding of diverse agonists converges to trigger outward movement of the cytoplasmic end of TM6 and inward movement of TM7 toward TM3 (*Rasmussen et al., 2011a*; *Nygaard et al., 2009*; *Venkatakrishnan et al., 2016*), thereby creating an intracellular crevice for G protein coupling (*Figure 5b*).

It is noteworthy that the common activation pathway we discovered in this study is not the only pathway that connecting extracellular ligand-binding and intracellular effector coupling for class A GPCRs – it is likely to be a shared portion of various activation pathways of GPCR members belonging to this class – each receptor still has its unique receptor-, ligand- and effector-specific activation pathways. In fact, research on this subject has boosted the discovery of selective and biased ligands (*McCorvy et al., 2018*; *Onaran et al., 2014*; *Schmid et al., 2017*; *Smith et al., 2018*; *Tan et al., 2018*; *Whalen et al., 2011*; *Wootten et al., 2018*; *Wootten et al., 2016*).

As shown in *Figure 5—figure supplement 2*, collapse of the Na$^+$ pocket leads to a denser repacking of six residues (five residue pairs), reflected by higher RRCS$_{sodium\_pocket}$ scores in active state than that in inactive state structures. Recently, the crystal structures (5X33 [*Hori et al., 2018*], 6BQG [*Peng et al., 2018*] and 6K1Q [*Nagiri et al., 2019*]) whose ligand (inverse agonist) diffuses deep in ligand-binding pocket or even occupies the sodium binding pocket (below D$^{2\times50}$) were reported. These inverse agonists disrupt the collapse of Na$^+$ pocket by blocking the rotation of W$^{6\times48}$ and/or taking the space of Na$^+$, and stabilize the receptors in an inactive state. Indeed, these inactive state structures showed zero RRCS$_{TM3-TM7}$ but high RRCS$_{TM3-TM6}$ scores. The inverse agonism are not only consistent with both our activation model and mutagenesis experiments, but also supported by the NMR study of A$_{2A}$R (*Eddy et al., 2018*). This study demonstrated the role of

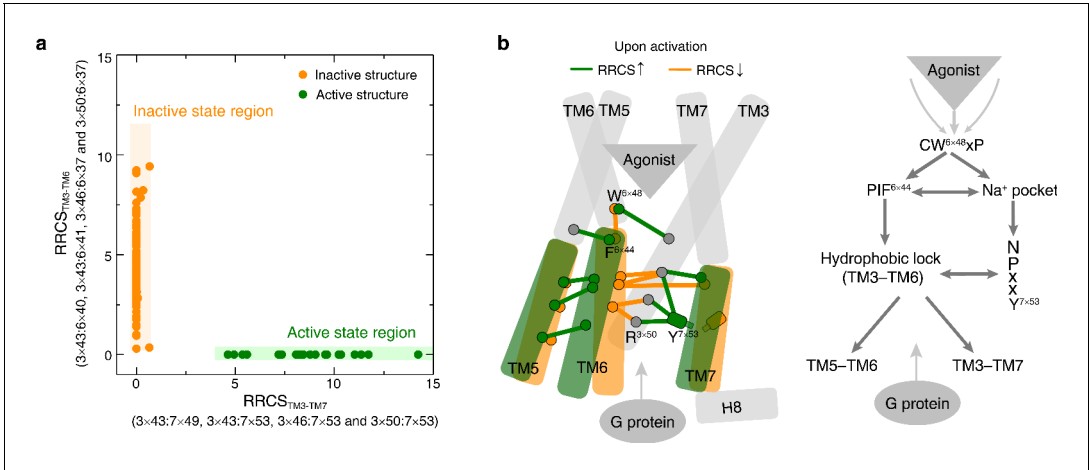

**Figure 5.** Common activation model of class A GPCRs reveals major changes upon GPCR activation. (a) Active and inactive state structures form compact clusters in the 2D inter-helical contact space: RRCS$_{TM3-TM7}$ (X-axis) and RRCS$_{TM3-TM6}$ (Y-axis). GPCR activation is best described by the outward movement of TM6 and inward movement of TM7, resulting in switch in the contacts of TM3 from TM6 to TM7. (b) Common activation model for class A GPCRs. Residues are shown in circles, conserved contact rearrangements of residue pairs upon activation are denoted by lines.

The online version of this article includes the following figure supplement(s) for figure 5:

**Figure supplement 1.** Global conformational change upon activation.

**Figure supplement 2.** An inverse-agonism of class A GPCRs by preventing the collapse of Na$^+$ pocket.

$D52^{2\times50}$ as an allosteric link between the orthosteric ligand-binding site and the intracellular signaling surface, revealing strong interactions with the toggle switch $W246^{6\times48}$.

## Experimental validation of the modular nature of the common activation pathway

Based on the knowledge of the common activation pathway, one would expect that mutations of residues in the pathway are likely to severely affect receptor activation. The two extreme consequences are constitutive activation (without agonist binding) or inactivation (abolished signalling). To experimentally test this hypothesis, we systematically designed site-directed mutagenesis for residues in the pathway on a prototypical receptor $A_{2A}R$, aiming to create constitutively activating/inactivating mutations (CAM/CIM), by promoting/blocking residue and helix level conformational changes revealed in the pathway. 6/15 designed CAMs and 15/20 designed CIMs were validated by functional cAMP accumulation assays, and none of them were reported before for $A_{2A}R$ (*Figure 6*, *Figure 6—figure supplement 1* and *Figure 6—source data 1*). The design of functional active/inactive mutants has been very challenging. However, the knowledge of common activation pathway of GPCRs presented here greatly improves the success rate. The mechanistic interpretation of 21 successful predicted mutants is explained below. We also discussed the 14 unsuccessful predictions in *Figure 6—source data 2*. Besides, we extended mutagenesis studies to $G_s$-coupled 5-HT$_7$ and $G_i$-coupled 5-HT$_{1B}$ receptors by designing CAM/CIMs in residues at $3\times40$, $3\times43$, $6\times40$, $6\times44$, and $7\times49$ (*Figure 6—figure supplement 2*).

In layer 1, the mutation $I92^{3\times40}N$ likely stabilizes the active state by forming amide-π interactions with $W246^{6\times48}$ and hydrogen bond with the backbone of $C185^{5\times461}$, which rewires the packing at the transmission switch and initiates the outward movement of the cytoplasmic end of TM6; this mutation elevated the basal cAMP level by sevenfold. Conversely, $I92^{3\times40}A$ would reduce the favorable contacts with $W^{6\times48}$ upon activation, which retards the initiation of the outward movement of TM6; this mutation resulted in a decrease in both basal cAMP level [71% of wild-type (WT)] and agonist potency (eightfold). Another example is the residue at $6\times44$, the mutation $F242^{6\times44}R$ would stabilize the inactive state by forming salt bridge with $D52^{2\times50}$, which blocks the rotation of TM6 and thus abolishes $G_s$ coupling; indeed this mutation greatly reduced basal cAMP level (to 63% WT) and agonist potency (by 374-fold). In contrast, $F242^{6\times44}A$ would reduce contacts with $W246^{6\times48}$, loosen TM3-TM6 contacts, diminish the energy barrier of TM6 release and make outward movement of TM6 easier; consistently this mutation elevated the basal cAMP level (by twofold) and increased the agonist potency (by eightfold). Mutations of residues forming the Na$^+$ pocket, such as $D52^{2\times50}A$ and $N280^{7\times45}R$, would destroy the hydrogen bond network at the Na$^+$ pocket and retard the initiation of the inward movement of TM7. These mutations completely abolished agonist potency and greatly reduced the basal cAMP level (to 80% and 78% of WT, respectively).

In layer 2, the mutations $L95^{3\times43}A/R$ and $I238^{6\times40}Y$ would loosen the hydrophobic lock, weaken TM3-TM6 contacts, promote the outward movement of cytoplasmic end of TM6 and eventually make receptor constitutively active; this is reflected by remarkably high basal cAMP production (28-, 2- and 11-fold increase, respectively). Notably, mutations at/near the Na$^+$ pocket, $L48^{2\times46}R$ and $N284^{7\times49}K$, could lock the Na$^+$ pocket at inactive packing mode by introducing salt bridge with $D52^{2\times50}$, thus blocking the inward movement of TM7 toward TM3. As expected, these mutations completely abolished agonist potency. The CIMs at/near the Na$^+$ pocket (from both layers 1 and 2) reflect that the subtle inward movement of TM7 towards TM3 is essential for receptor activation, which is often underappreciated and overshadowed by the movement of TM6. In line with this, two mutations on TM7, $N284^{7\times49}A$ and $Y288^{7\times53}A$, attenuate the TM3-TM7 contacts upon activation and completely abolished or greatly reduced (by 16-fold) agonist potency, respectively.

In layer 3, $I98^{3\times46}A$ likely reduces contacts with $Y288^{7\times53}$, weakens the packing between TM3-TM7, and retards the movement of TM7 toward TM3; similarly, $L235^{6\times37}A$ would reduce contacts with $F201^{5\times62}$, weaken the packing between TM5-TM6, and makes the TM6 movement toward TM5 more difficult. In line with the interpretation, these mutations resulted in reduced basal cAMP level (72% and 71% WT, respectively) and decreased agonist potency (23- and 4-fold, respectively). These results are consistent with previous findings on vasopressin type-2 receptor (V2R) (*Venkatakrishnan et al., 2016*).

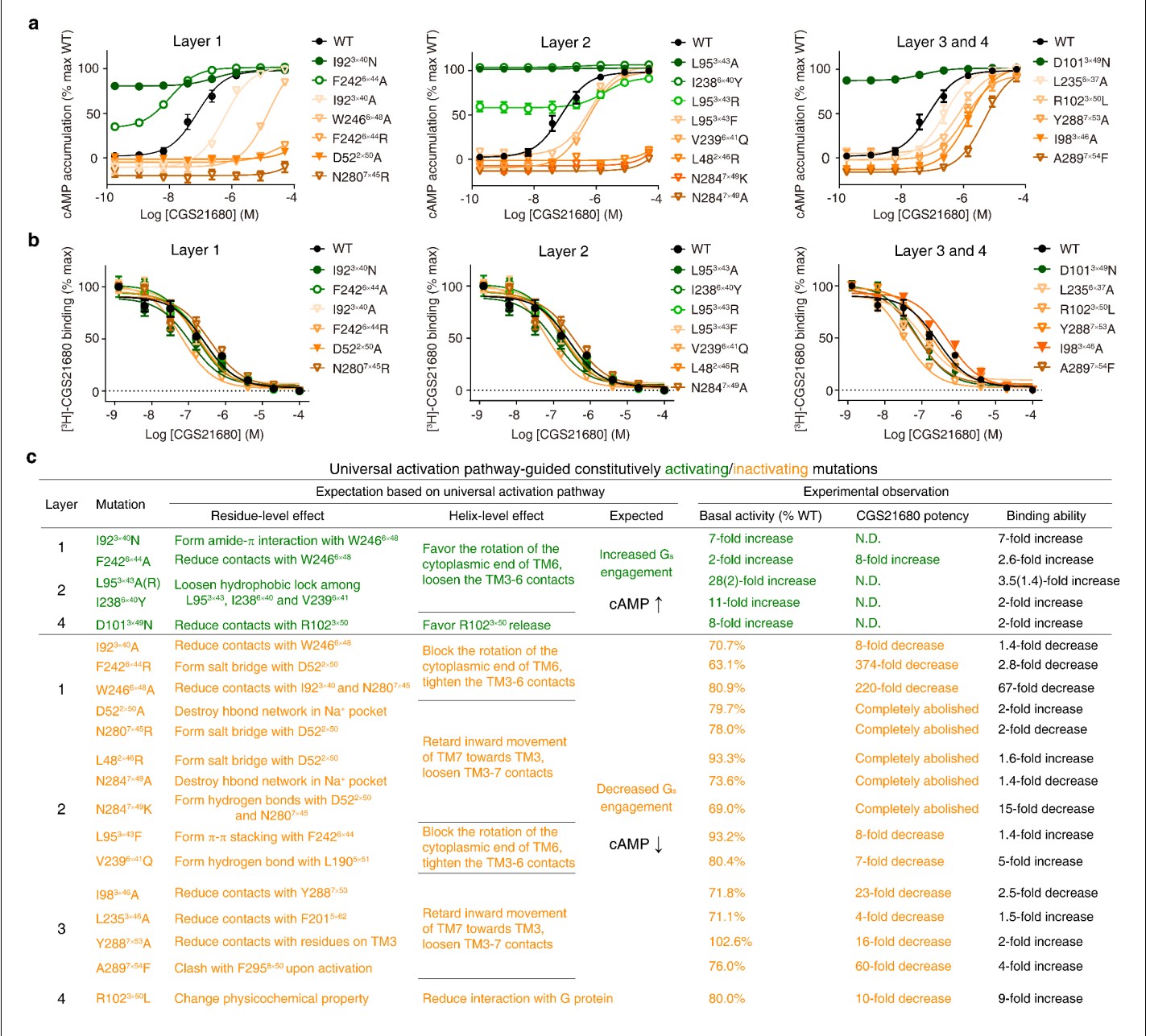

**Figure 6.** Experimental validation of the common activation mechanism. (**a**) cAMP accumulation assay and (**b**) radioligand binding assay: both validated the common activation pathway-guided design of CAMs/CIMs for A$_{2A}$R. Wildtype (WT), CAMs and CIMs are shown in black, green and orange, respectively. (**c**) Mechanistic interpretation of common activation pathway-guided CAMs/CIMs design. N.D.: basal activity was too high to determine an accurate EC$_{50}$ value.

The online version of this article includes the following source data and figure supplement(s) for figure 6:

**Source data 1.** Functional and ligand binding properties of A$_{2A}$R mutations.
**Source data 2.** Analysis of the 14 unsuccessful predictions of A$_{2A}$R CAMs/CIMs.
**Figure supplement 1.** Experimental validation of common activation pathway-guided CAM/CIM design for A 2A R.
**Figure supplement 2.** Experimental validation of common activation pathway-guided CAM/CIM design for G$_s$-coupled 5-HT$_7$ and G$_i$-coupled 5-HT$_{1B}$ receptors.

In layer 4, D101$^{3 \times 49}$N likely diminishes its intra-helical interaction with R102$^{3 \times 50}$ and thus makes the release of the latter easier, which in turn promotes G-protein recruitment. Consistent with this possibility, this mutation led to a greatly elevated basal cAMP level (eightfold).

Despite these A$_{2A}$R mutants greatly affect receptor activation, our radioligand binding assay shows that they generally retain the agonist binding ability, with the exception of two CIMs: W246$^{6 \times 48}$A and N284$^{7 \times 45}$K (*Figure 6b,c* and *Figure 6—source data 1*). This suggests that the common activation pathway is of modular nature and that such an organization allows for a significant number of residues involved in agonist binding to be uncoupled from receptor activation/inactivation and G-protein binding.

As shown in *Figure 6—figure supplements 2* and 5-HT$_7$ receptor mutations F336$^{6 \times 44}$R and N380$^{7 \times 49}$K completely abolished agonist potency and greatly reduced the basal cAMP level, which is remarkably consistent with the observation on A$_{2A}$R, highlighting the crucial role of the highly conserved residues F$^{6 \times 44}$ and N$^{7 \times 49}$. Beyond G$_s$-coupled A$_{2A}$R and 5-HT$_7$ receptor, we also validated this mutation design in G$_i$-coupled 5-HT$_{1B}$ receptor. Indeed, two CIMs, I137$^{3 \times 40}$N and F323$^{6 \times 44}$H greatly reduced receptor-mediated G$_i$ activity compared to WT, whereas three CAMs, L173$^{3 \times 43}$A in G$_s$-coupled 5-HT$_7$ receptor, F323$^{6 \times 44}$A and I137$^{3 \times 40}$A in G$_i$-coupled 5-HT$_{1B}$ receptor, were verified to promote their basal activities, consistent with the observation on CAMs (L95$^{3 \times 43}$A, F242$^{6 \times 44}$A and I92$^{3 \times 40}$A) designed for A$_{2A}$R.

## The common pathway allows mechanistic interpretation of mutations

Four hundred thirty five disease-associated mutations were collected, among which 28% can be mapped to the common activation pathway, much higher than that to the ligand-binding and G-protein-binding regions (20% and 7%, respectively) (*Figure 7a,b*). Furthermore, 272 CAMs/CIMs from 41 receptors (*Figure 7c*) were mined from the literature for the 14 hub residues (i.e. residues that have more than one edges in the pathway).

The average number of disease-associated mutations in the common activation pathway is much higher than that of ligand-binding pocket, G-protein-binding site, and residues in other regions (2.5-, 3.5- and 3.5-fold, respectively), reflecting the enrichment of disease-associated mutations on the pathway (*Figure 7a*). Within this pathway, the enrichment of disease mutations and CAMs/CIMs in layers 1 and 2 is noteworthy, which highlights the importance of signal initiation and hydrophobic lock opening, and further supports the modular and hierarchical nature of GPCR activation (*Figures 3* and *5b*). Notably, for certain residues, such as D$^{2 \times 50}$ and Y$^{7 \times 53}$, only loss-of-function disease mutations or CIMs were observed (*Figure 7*), implying they are indispensable for receptor activation and the essential role of TM7 movement (*Figures 3* and *5*).

The functional consequence of these single point mutations can be rationalized by analysing if they are stabilizing/destabilizing the contacts in the common activation pathway or promoting/retarding the required helix movement upon activation (*Figure 7b* and *Figure 7—figure supplement 1*). For example, I130$^{3 \times 43}$N/F (layer 2) in V2R was reported as a gain-/loss-of-function mutation that causes nephrogenic syndrome of inappropriate antidiuresis (*Erdélyi et al., 2015*) or nephrogenic diabetes insipidus (*Pasel, 2000*), respectively. I130$^{3 \times 43}$N/F likely loosens/stabilizes the hydrophobic lock, weakens/strengthens the TM3-TM6 packing and leads to constitutively active/inactive receptors. Another example is T58$^{1 \times 53}$M in rhodopsin, which was reported as a loss-of-function mutation that causes retinitis pigmentosa 4 (*Napier et al., 2015*). T58$^{1 \times 53}$M likely increases hydrophobic contacts with Y306$^{7 \times 53}$ and P303$^{7 \times 50}$, which retards the inward movement of TM7 towards TM3 and eventually decreases G-protein recruitment. As in the case of disease-associated mutations, CAMs/CIMs that have been previously reported in the literature can also be interpreted by the framework of common activation pathway (*Figure 7—figure supplement 1b*). For example, F248$^{6 \times 44}$Y in CXCR4 (*Wescott et al., 2016*) was reported as a CIM. This residue likely forms hydrogen bond with S123$^{3 \times 39}$, which blocks the rotation of the cytoplasmic end of TM6, and decreases G-protein engagement.

Not surprisingly, the 35 residues constituting the pathway are highly conserved across class A GPCRs, dominated by physiochemically similar amino acids (*Figure 7—figure supplement 2*). The average sequence similarity of these positions across 286 non-olfactory class A receptors is 66.2%, significantly higher than that of ligand-binding pockets (31.9%) or signaling protein-coupling regions (35.1%). Together, these observations suggest that the modular and hierarchical nature of the activation pathway allows decoupling of the ligand-binding pocket, G-protein-binding site and the

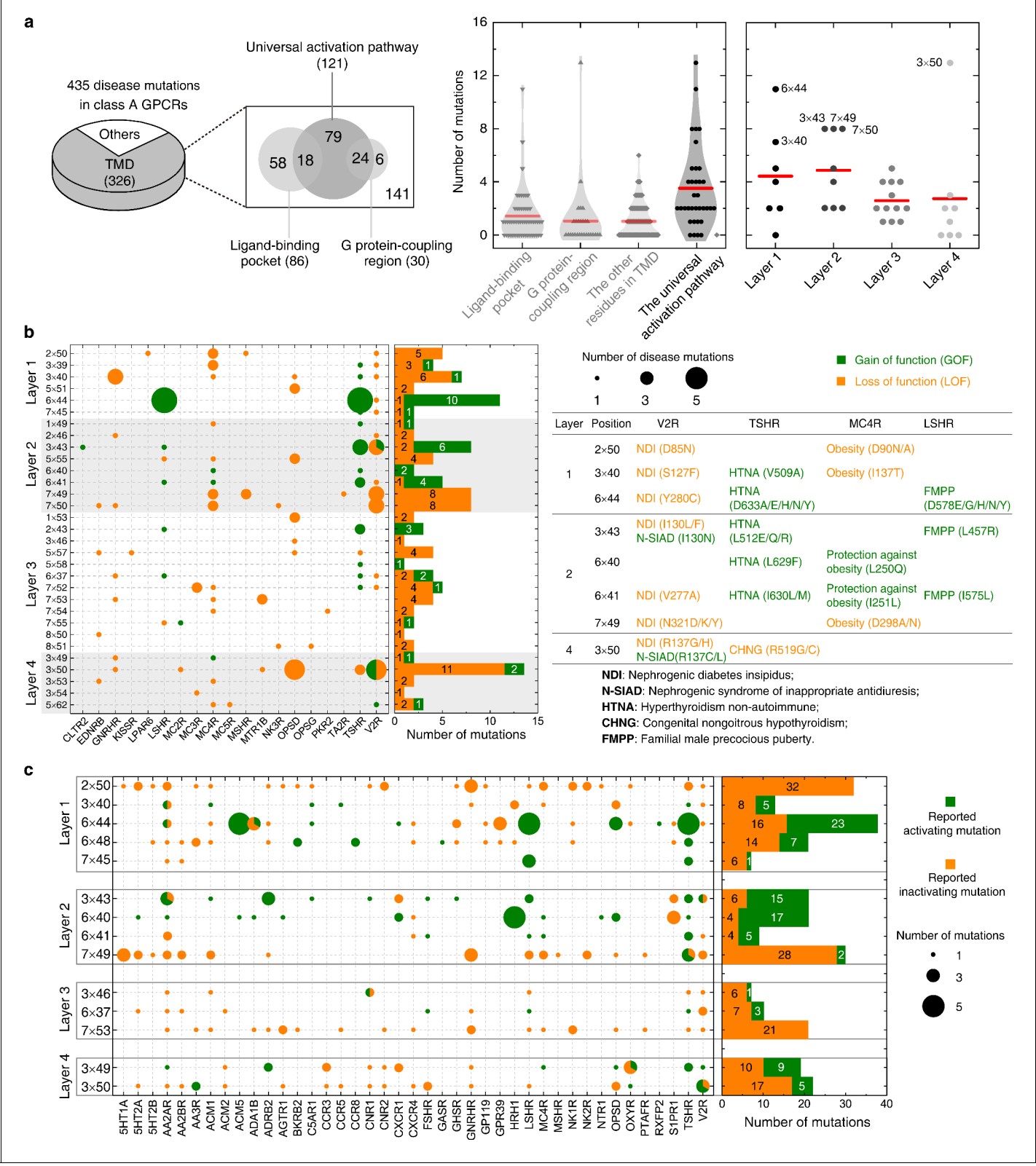

**Figure 7.** Importance of the common activation pathway in pathophysiological and biological contexts. (**a**) Comparison of disease-associated mutations in the common activation pathway (further decomposed into layers 1–4), ligand-binding pocket, G-protein-coupling region and other regions. Red line denotes the mean value. (**b**) Mapping of disease-associated mutations in class A GPCRs to the common activation pathway. (**c**) Key roles of the residues

*Figure 7 continued on next page*

*Figure 7 continued*

constituting the common activation pathway have been reported in numerous experimental studies on class A GPCRs. Two hundred seventy two (272) CAMs/CIMs from 41 receptors were mined from the literature for the 14 hub residues (i.e. residues that have more than one edges in the pathway). The online version of this article includes the following source data and figure supplement(s) for figure 7:

**Source data 1.** Constitutively activating/inactivating mutations for the 14 hub residues in the common activation pathway.
**Figure supplement 1.** The common activation pathway can be used to mechanistically interpret disease-associated mutations and CAMs/CIMs.
**Figure supplement 2.** Residues in the common activation pathway are more conserved than other functional regions of GPCR.

residues contributing to the common activation mechanism. Such an organization of the receptor might facilitate the uneven sequence conservation between different regions of GPCRs, confers their functional diversity in ligand recognition and G-protein binding while still retaining a common activation mechanism.

## Discussion

Using a novel, quantitative residue contact descriptor, RRCS, and a family-wide comparison across 234 structures from 45 class A GPCRs, we reveal a common activation pathway that directly links ligand-binding pocket and G-protein-coupling region. Key residues that connect the different modules allows for the decoupling of a large number of residues in the ligand-binding site, G-protein contacting region and residues involved in the activation pathway. Such an organization may have facilitated the rapid expansion of GPCRs through duplication and divergence, allowing them to evolve independently and bind to diverse ligands due to removal of the constraint (*i.e.*, between a large number of ligand-binding residues and those involved in receptor activation). This model unifies many previous important motifs and observations on GPCR activation in the literature (CWxP, PIF, Na$^+$ pocket, NPxxY, DRY and hydrophobic lock) and is consistent with numerous experimental findings.

We focused on the common activation pathway (i.e. the common part of activation mechanism shared by all class A GPCRs and various intracellular effectors) in this study. Obviously, individual class A receptor naturally has its intrinsic activation mechanism(s), as a result of the diversified sequences, ligands and physiological functions. Indeed, receptor-specific activation pathways (including mechanisms of orthosteric, positive or negative allosteric modulators, biased signaling/selectivity of downstream effectors) have been revealed by both experimental studies including biophysical (such as X-ray, cryo-EM, NMR, FRET/BRET and DEER), biochemical and computational approaches (such as evolutionary trace analysis and molecular dynamics simulations), especially for the prototypical receptors such as rhodopsin, $\beta_2$-adrenergic and A$_{2A}$ receptors. These studies demonstrated the complexity and plasticity of signal transduction of GPCRs. The computational framework we have developed may assist us in better understanding the mechanism of allosteric modulation, G-protein selectivity and diverse activation processes via intermediate states as more GPCR structures become available. While we interpret the changes as a linear set of events, future studies aiming at understanding dynamics could provide further insights into how the common activation mechanism operates in individual receptors.

Given the common nature of this pathway, we envision that the knowledge obtained from this study can not only be used to mechanistically interpret the effect of mutations in biological and pathophysiological context but also to rationally introduce mutations in other receptors by promoting/blocking residue and helix level movements that are essential for activation. Such protein engineering approaches may enable us to make receptors in specific conformational states to accelerate structure determination studies using X-ray crystallography or electron microscopy and functional investigation in the future. The method developed here could also be readily adapted to map allosteric pathways and reveal mechanisms of action for other key biological systems such as kinases, ion channels and transcription factors.

# Materials and methods

## Glossary

Transmembrane domains (TMD): the core domain exists in all GPCRs, and consists of seven-trans-membrane helices (TM1–7) that are linked by three extracellular loops (ECL1-3) and three intracellular loops (ICL1-3).

GPCRdb numbering scheme: a structure-based numbering system for GPCRs (*Isberg et al., 2016*; *Isberg et al., 2015*), an improved version of sequence-based Ballesteros–Weinstein numbering (*Ballesteros and Weinstein, 1995*) that considers structural distortions such as helical bulges or constrictions. The most conserved residue in a helix n is designated n×50, while other residues on the helix are numbered relative to this position.

Node: a point in a network at which lines intersect, branch or terminate. In this case, nodes represent amino acid residues.

Edge: a connection between the nodes in a network. In this case, an edge represents a residue-residue contact.

Hub: a node with two or more edges in a network.

Constitutively activating mutation (CAM): a mutant that could increase the inherent basal activity of the receptor by activating the G-protein-signaling cascade in the absence of agonist.

Constitutively inactivating mutation (CIM): a mutant completely abolishes receptor signalling.

## GPCR structure data set

As of October 1, 2018, there are 234 released structures of 45 class A GPCRs with resolution better than 3.8 Å (*Figure 1—source data 1*), which covers 71% (203 out of 286 receptors, including 158 receptors that have no structures but share >50% sequence similarity in the TMD with the 45 structure-determined receptors) of class A GPCRs (*Figure 1a*). Based on the type of bound ligand and effector, these structures could be classified into three states: inactive state (antagonist or inverse agonist-bound, 142 structures from 38 receptors), active state (both agonist- and G protein/G protein mimetic-bound, 27 structures from eight receptors) and intermediate state (only agonist-bound, 65 structures from 15 receptors). In this study, we primarily focused on conformational comparison between inactive- and active- state structures, while also investigating the intermediate state structures. In the structure data set, seven receptors have both inactive and active structures: rhodopsin (bRho), $\beta_2$-adrenergic receptor ($\beta_2$AR), M2 muscarinic receptor (M2R), μ-opioid receptor (μOR), adenosine $A_{2A}$ receptor ($A_{2A}$R), κ-opioid receptor (κ-OR) and adenosine $A_{1A}$ receptor ($A_{1A}$R), the active state structure of which was recently determined by cryo-EM. In addition, 32 receptors have either inactive or active structures (*Figure 1—source data 1*).

## Calculation of residue-residue contact score (RRCS)

We developed a much finer distance-based method (than coarse-grained Boolean descriptors such as contact map and residues contact [*Venkatakrishnan et al., 2016*; *Kayikci et al., 2018*; *Eldridge et al., 1997*; *Verdonk et al., 2003*; *Wang et al., 2017*; *Adhikari and Cheng, 2016*]), namely residue-residue contact score (RRCS). For a pair of residues, RRCS is calculated by summing up a plateau-linear-plateau form atomic contact score adopted from GPCR–CoINPocket (*Ngo et al., 2017*; *Kufareva et al., 2011*; *Kufareva and Abagyan, 2012*; *Kufareva et al., 2014a*; *Kufareva et al., 2013*; *Kufareva et al., 2014b*; *Marsden and Abagyan, 2004*) for each possible inter-residue heavy atom pairs (*Figure 2—figure supplement 1a*). GPCR–CoINPocket is a modified version of the hydrophobic term of ChemScore (*Eldridge et al., 1997*; *Verdonk et al., 2003*) that has been successfully used to describe hydrophobic contribution to binding free energy between ligand and protein. RRCS can describe the strength of residue-residue contact quantitatively in a much more accurate manner than Boolean descriptors (*Venkatakrishnan et al., 2016*; *Flock et al., 2017*). For example, Boolean descriptors do not capture side chain repacking if the backbone atoms of the two residues are close to each other (e.g. translocation of $Y^{7\times53}$ away from residue at 2×43 upon GPCR activation) and local contacts involving adjacent residues (residues within four/six amino acids in protein sequence) (e.g., disengagement between $D/E^{3\times49}$ and $R^{3\times50}$), while both cases can be well reflected by the change of RRCS (*Figure 2c* and *Figure 2—figure supplement 1b*).

All RRCS data can be found in *Figure 2—source data 1*. The computational details are described as below:

1. For the residue pairs between adjacent residues that are within four amino acids in protein sequence, only side chain heavy atom pairs were considered, atom pairs involving in backbone atoms (Cα, C, O, N) were excluded, since the latter seldom change during GPCR activation. For other residue pairs, all possible heavy atom pairs (including backbone atoms) were included when calculating RRCS.

2. Atomic contact scores are solely based on interatomic distance, and they are treated equally without weighting factors such as atom type or contact orientation. In principle, weighting of atomic contact by atom type and/or orientation would improve residue-residue contact score. However, parameterization of atom type or contact orientation is relatively arbitrary, subjective and complicated, especially considering the lipid bilayer environment surrounding GPCRs. Our preliminary study for 12 structures from six receptors (bRho, β$_2$AR, M2R, μOR, A$_{2A}$R and κ-OR) revealed that amino acids with hydrophobic side chains (one-letter code: A, V, I, L, M, P, F, Y, W) contribute to the majority (~88%) of residue pairs. Meanwhile, ionic lock opening of well-known motif DRY upon receptor activation can be adequately reflected by RRCS change between D/E$^{3×49}$ and R$^{3×50}$. These results suggest that interatomic distance-dependent residue pair contact score may represent an acceptable approximation of actual (either hydrophobic or charge-charge) interaction energies (*Ngo et al., 2017*) and is accurate enough for identifying conserved rearrangements of residue contacts upon receptor activation.

3. The quality of structures is extremely important for RRCS calculation. We adopted two criteria to exclude unreliable structures and residues: (a) crystal structures whose resolution is ≥3.8 Å. Structures in this category are: 5DGY (7.70 Å), 2I37 (4.20 Å), 2I36 (4.10 Å), 5TE5 (4.00 Å), 4GBR (4.00 Å), 5NJ6 (4.00 Å), 5V54 (3.90 Å), 2I35 (3.80 Å), 5D5B (3.80 Å), 4XT3 (3.80 Å); (b) residues whose residue-based real-space R-value (RSR [*Jones et al., 1991*]) is greater than 0.35. RSR is measure of how well 'observed' and calculated electron densities agree for a residue. RSR ranges from 0 (perfect match) to 1 (no match); RSR greater than 0.4 indicates a poor fit (*Smart et al., 2018*). Here we adopted a stricter cut-off, 0.35. Among the 234 class A GPCR structures, 156 have available RSR information (*Kleywegt et al., 2004*) (http://eds.bmc.uu.se), with 8.8% residues have RSR >0.35 and they are omitted in our analysis. For the 35 residues that constitute the common activation pathway, 255 out of 5460 RSR data points (~4.7%, lower than 8.8% for all residues) were omitted for having RSR values > 0.35.

4. For structures with multiple chains, RRCS were the average over all chains. For residues with multiple alternative conformations, RRCS was the sum of individual values multiplied by the weighting factor: occupancy value extracted from PDB files. Small molecule/peptide ligand, or intracellular binding partner (G protein or its mimetic) was treated as a single residue.

5. For the family-wide comparison of conformational changes upon activation, structurally equivalent residues are numbered by GPCRdb numbering scheme (*Isberg et al., 2016*; *Isberg et al., 2015*). Of the 35 residues in the common activation pathway, their GPCRdb numbering in all structures is almost identical to the Ballesteros–Weinstein numbering (*Ballesteros and Weinstein, 1995*), the exceptions are residues at 6×37, 6×41 and 6×44 for five receptors: FFAR1, P2Y$_1$, P2Y$_{12}$, F2R and PAR2, which are all from the delta branch of class A family.

## Identification of conserved rearrangements of residue contacts upon activation

Using RRCS, structural information of TMD and helix eight in each structure can be decomposed into 400 ~ 500 residue pairs with positive RRCS. ΔRRCS, defined as RRCS$_{active}$ − RRCS$_{inactive}$, reflects the change of RRCS for a residue pair from inactive- to active- state (*Figure 2b–d* and *Figure 2—figure supplement 1b*). To identify residue pairs with conserved conformational rearrangements upon activation across class A GPCRs, two rounds of selections (*Figure 2d* and *Figure 2—source data 1*) were performed: (i) identification of conserved rearrangements of residue contacts upon activation for six receptors (bRho, β$_2$AR, M2R, μOR, A$_{2A}$R and κ-OR), that is equivalent residue pairs show a similar and substantial change in RRCS between the active and inactive state structure of each of the six receptors (the same sign of ΔRRCS and |ΔRRCS| > 0.2 for all receptors) and (ii) family-wide RRCS comparison between the 142 inactive and 27 active state structures to identify residues pairs of statistically significant different (p<0.001; two sample *t*-test) RRCS upon activation.

Round 1. Identification of conserved rearrangements of residue contacts. Six receptors with available inactive- and active- state structures were analyzed using ΔRRCS to identify residue pairs that

share similar conformational changes. Twelve representative crystal structures (high-resolution, no mutation or one mutation in TMD without affecting receptor signalling) were chosen in this stage: six inactive state structures (PDB codes 1GZM for bRho, 2RH1 for $\beta_2$AR, 3UON for M2R, 4DKL for $\mu$OR, 3EML for $A_{2A}$R and 4DJH for $\kappa$-OR) and six active state structures (3PQR for bRho, 3SN6 for $\beta_2$AR, 4MQS for M2R, 5C1M for $\mu$OR, 5G53 for $A_{2A}$R and 6B73 for $\kappa$-OR) (*Figure 2d*, *Figure 2—figure supplement 1c* and *Figure 2—source data 1*). Each receptor has approximately 600 residues pairs that have positive RRCS. Roughly one quarter are newly formed during receptor activation ($RRCS_{inactive} = 0$ and $RRCS_{active} > 0$); another quarter lose their contacts upon receptor activation ($RRCS_{inactive} > 0$ and $RRCS_{active} = 0$); and the remaining appear in both the inactive- or active- state structures ($RRCS_{inactive} > 0$ and $RRCS_{active} > 0$), the contact rearrangement of which can only be reflected by $\Delta$RRCS, but not Boolean descriptors.

To identify residue pairs that share conserved rearrangements of residue contacts upon activation, two steps are performed to qualify residue pairs for the next round. Firstly, residue pairs with same sign of $\Delta$RRCS and $|\Delta RRCS| > 0.2$ for all six receptors were identified. There are 32 intra-receptor residues pairs ($1\times49{:}7\times50$, $1\times53{:}7\times53$, $1\times53{:}7\times54$, $2\times37{:}2\times40$, $2\times42{:}4\times45$, $2\times43{:}7\times53$, $2\times45{:}4\times50$, $2\times46{:}2\times50$, $2\times50{:}3\times39$, $2\times57{:}7\times42$, $3\times40{:}6\times48$, $3\times43{:}6\times40$, $3\times43{:}6\times41$, $3\times43{:}7\times49$, $3\times43{:}7\times53$, $3\times46{:}6\times37$, $3\times46{:}7\times53$, $3\times49{:}3\times50$, $3\times50{:}3\times53$, $3\times50{:}6\times37$, $350{:}7\times53$, $3\times51{:}5\times57$, $5\times51{:}6\times44$, $5\times58{:}6\times40$, $5\times62{:}6\times37$, $6\times40{:}7\times49$, $6\times44{:}6\times48$, $7\times50{:}7\times55$, $7\times52{:}7\times53$, $7\times53{:}8\times50$, $7\times54{:}8\times50$ and $7\times54{:}8\times51$) and five receptor-G protein/its mimetic residue pairs ($3\times50{:}G$ protein, $3\times53{:}G$ protein, $3\times54{:}G$ protein, $5\times61{:}G$ protein and $6\times33{:}G$ protein) that meet this criterion. Secondly, we also investigated residue pairs with $\Delta$RRCS that are conserved in five receptors (*i.e.*, with one receptor as exception). Considering there is no $Na^+$ pocket for rhodopsin, three residue pairs ($2\times50{:}7\times49$, $6\times44{:}7\times45$ and $6\times48{:}7\times45$) around $Na^+$ pocket were analyzed for five receptors but not bRho. Additionally, three residue pairs have 0 ($3\times46{:}3\times50$, $5\times55{:}6\times41$) or negative ($7\times45{:}7\times49$) $\Delta$RRCS for $\kappa$-OR but positive $\Delta$RRCS for the other five receptors. As for $3\times46{:}3\times50$, nanobody-stabilized active structures ($\beta_2$AR: 3P0G, 4LDO, 4LDL, 4LDE, 4QKX; and $\mu$OR: 5C1M) generally have lower contact scores (<0.4) compared with G-protein-bound active-state structures (2.17 for 3SN6 of $\beta_2$AR, 2.57 for 5G53 of $A_{2A}$R and 6.93 for 3PQR of bRho). For these residue pairs, we added newly determined $G_i$-bound active $A_{1A}$R and 5-HT$_{1B}$ receptor and found that they have positive $\Delta$RRCS, like other five receptors (*Figure 4—figure supplements 1* and *2*). Thus, these three residue pairs ($3\times46{:}3\times50$, $5\times55{:}6\times41$ and $7\times45{:}7\times49$) were retained. Totally, six residue pairs with conserved $\Delta$RRCS in five receptors were rescued. Taken together, 38 intra-receptor residue pairs and five receptor-G protein/its mimetic residue pairs were identified to have conserved rearrangements of residue contacts upon activation.

Round 2. Family-wide conservation analysis of residue contact pattern. To investigate the conservation of residue contact pattern for the 38 intra-receptor residue pairs across these functionally diverse receptors, two-tailed unpaired *t*-test between inactive state (142 inactive structures from 38 receptors) and active state (27 active structures from eight receptors) groups were performed (*Figure 2d* and *Figure 2—source data 2*). Thirty one residue pairs have significantly different RRCS between inactive- and active-state ($p<10^{-5}$). As rhodopsin lacks the $Na^+$ pocket, all rhodopsin structures were neglected in the analysis of 3 residue pairs around the pocket ($2\times50{:}7\times49$, $6\times44{:}7\times45$ and $6\times48{:}7\times45$), which have good p value ($<10^{-3}$) for these non-rhodopsin class A GPCRs. Four residue pairs were filtered out in this round due to their poor p value, that is there are no statistically significant difference in RRCS between inactive and active states (p=0.01 for $2\times37{:}2\times40$, 0.96 for $2\times42{:}4\times45$, 0.02 for $2\times45{:}4\times50$ and 0.014 for $2\times57{:}7\times42$).

Finally, 34 intra-receptor residue pairs (*Figure 2d*, *Figure 4—figure supplements 1* and *2*) and five receptor-G-protein residue pairs were identified with conserved rearrangements of residue contacts upon activation, including all six residues pairs identified by the previous RC approaches (*Venkatakrishnan et al., 2016*).

## Sequence analysis of class A GPCRs

The alignment of 286 non-olfactory, class A human GPCRs were obtained from the GPCRdb (*Isberg et al., 2016*; *Isberg et al., 2015*). The distribution of sequence similarity/identity across class A GPCRs were extracted from the sequence similarity/identity matrix for different structural regions using 'Similarity Matrix' tool in GPCRdb. The sequence conservation score (*Figure 1—figure supplement 1*) for all residue positions across 286 non-olfactory class A GPCRs were evaluated by the

Protein Residue Conservation Prediction (*Capra and Singh, 2007*) tool with scoring method 'Property Entropy' (*Mirny and Shakhnovich, 1999*). Sequence conservation analysis (*Figure 7—figure supplement 2*) were visualized by WebLogo3 (*Crooks et al., 2004*) with sequence alignment files from GPCRdb as the input.

## CAM/CIM in class A GPCRs

For the 14 hub residues in the common activation pathway, we collected the functional mutation data from the literature and GPCRdb (*Isberg et al., 2016*; *Isberg et al., 2015*). Mutations with 'more than two fold-increase in basal activity/constitutively active' or 'abolished effect' compared to the wild-type receptor were selected. Together, 272 mutations from 41 class A GPCRs on the 14 hub residues were collected, including the mutations we designed and validated in this study (*Figure 7—source data 1*).

## Disease-associated mutations in class A GPCRs

To reveal the relationship between disease-associated mutations and related phenotypes of different transmembrane regions (*Vassart and Costagliola, 2011*; *Thompson et al., 2014*; *Tao, 2006*; *Tao, 2008*), we collected disease-associated mutation information for all 286 non-olfactory class A GPCRs by database integration and literature investigation. Four commonly used databases (UniProt [*The UniProt Consortium, 2017*], OMIM [*Amberger et al., 2011*], Ensembl [*Zerbino et al., 2018*] and GPCRdb [*Isberg et al., 2016*; *Isberg et al., 2015*]) were first filtered by disease mutations and then merged. Totally 435 disease mutations from 61 class A GPCRs were collected (*Figure 1—source data 2*).

## Pathway-guided CAM/CIM design in $A_{2A}R$

We designed mutations for a prototypical receptor $A_{2A}R$, guided by the common activation pathway, aiming to get constitutively active/inactive receptor. Mutations that can either stabilize active or inactive state structures of $A_{2A}R$ or promote/block conformational changes upon activation were designed (*Figure 6c* and *Figure 6—figure supplement 1*) and tested by a functional cAMP accumulation assay. The inactive state structure 3EML and active state structure 5G53 were used. In silico mutagenesis was performed by Residue Scanning module in BioLuminate (*Beard et al., 2013*). Side-chain prediction with backbone sampling and a cut-off value of 6 Å were applied during the scanning. ΔStability is the change of receptor stability when introducing a mutation. We filtered the mutations by one of the following criteria: (i) ΔStability in active and inactive structures have opposite signs; or (ii) ΔStability in active and inactive structures have the same sign, but favorable interactions such as hydrogen bonds, salt bridge or pi-pi stacking exist in only one structure that can promote/ block conformational changes upon activation. Totally, 15 and 20 mutations were predicted to be CAMs and CIMs, respectively. (*Figure 6c* and *Figure 6—figure supplement 1*).

## cAMP accumulation assays

(i)$A_{2A}R$. The desired mutations were introduced into amino-terminally Flag tag-labeled human $A_{2A}R$ in the pcDNA3.1 vector (Invitrogen, Carlsbad, CA). This construct displayed equivalent pharmacological features to that of untagged human receptor based on radioligand binding and cAMP assays (*Massink et al., 2015*). The mutants were constructed by PCR-based site-directed mutagenesis (Muta-directTM kit, Beijing SBS Genetech Co., Ltd., China). Sequences of receptor clones were confirmed by DNA sequencing. HEK-293 cells (obtained from ATCC and confirmed as negative for mycoplasma contamination) were seeded onto 6-well cell culture plates. After overnight culture, the cells were transiently transfected with WT or mutant DNA using Lipofectamine 2000 transfection reagent (Invitrogen). After 24 hr, the transfected cells were seeded onto 384-well plates (3,000 cells per well). cAMP accumulation was measured using the LANCE cAMP kit (PerkinElmer, Boston, MA) according to the manufacturer's instructions. Briefly, transfected cells were incubated for 40 min in assay buffer (DMEM, 1 mM 3-isobutyl-1-methylxanthine) with different concentrations of agonist [CGS21680 (179 pM to 50 μM)]. The reactions were stopped by addition of lysis buffer containing LANCE reagents. Plates were then incubated for 60 min at room temperature and time-resolved FRET signals were measured at 625 nm and 665 nm by an EnVision multilabel plate reader (PerkinElmer). The cAMP response is depicted relative to the maximal response of CGS21680 (100%) at

the WT $A_{2A}R$. (ii) 5-HT$_{1B}$ receptor. cAMP accumulation was measured using LANCE cAMP kit (PerkinElmer). Briefly, HEK293T (obtained from and certified by the Cell Bank at the Chinese Academy of Science and confirmed as negative for mycoplasma contamination) cells were transfected with plasmids bearing WT or mutant 5-HT$_{1B}$ receptor. Cells were collected 24 hr post-transfection and used to seed white poly-D-lysine coated 384-well plates at a density of 2,000 cells per well. Cells were incubated for a further 24 hr at 37˚C. Cells were then incubated for 30 min in assay buffer (HBSS, 5 mM HEPES, 0.1% BSA, 0.5 mM 3-isobutyl-1-methylxanthine) with constant Forsklin (800 nM) and different concentrations of dihydroergotamine (DHE, 0.64 pM to 50 nM) at 37˚C. The reactions were stopped by addition of lysis buffer containing LANCE reagents. Plates were then incubated for 60 min at room temperature, and time-resolved FRET signals were measured after excitation at 620 nm and 650 nm by EnVision (PerkinElmer).

## CGS21680 binding assay

CGS21680 (a specific adenosine $A_{2A}$ subtype receptor agonist) binding was analyzed using plasma membranes prepared from HEK-293 cells transiently expressing WT and mutant $A_{2A}Rs$. Approximately $1.2 \times 10^8$ transfected HEK-293 cells were harvested, suspended in 10 ml ice-cold membrane buffer (50 mM Tris-HCl, pH 7.4) and centrifuged for 5 min at 700 **g**. The resulting pellet was resuspended in ice-cold membrane buffer, homogenized by Dounce Homogenizer (Wheaton, Millville, NJ) and centrifuged for 20 min at 50,000 **g**. The pellet was resuspended, homogenized, centrifuged again and the precipitate containing the plasma membranes was then suspended in the membrane buffer containing protease inhibitor (Sigma-Aldrich, St. Louis, MO) and stored at −80˚C. Protein concentration was determined using a protein BCA assay kit (Pierce Biotechnology, Pittsburgh, PA). For homogeneous binding, cell membrane homogenates (10 µg protein per well) were incubated in membrane binding buffer (50 mM Tris-HCl, 10 mM NaCl, 0.1 mM EDTA, pH 7.4) with constant concentration of [$^3$H]-CGS21680 (1 nM, PerkinElmer) and serial dilutions of unlabeled CGS21680 (0.26 nM to 100 µM) at room temperature for 3 hr. Nonspecific binding was determined in the presence of 100 µM CGS21680. Following incubation, the samples were filtered rapidly in vacuum through glass fiber filter plates (PerkinElmer). After soaking and rinsing four times with ice-cold PBS, the filters were dried and counted for radioactivity in a MicroBeta2 scintillation counter (PerkinElmer).

## Surface expression of $A_{2A}Rs$

HEK293 cells were seeded into six-well plate and incubated overnight. After transient transfection with WT or mutant plasmids for 24 hr, the cells were collected and blocked with 5% BSA in PBS at room temperature for 15 min and incubated with primary anti-Flag antibody (1:100, Sigma-Aldrich) at room temperature for 1 hr. The cells were then washed three times with PBS containing 1% BSA followed by 1 hr incubation with anti-rabbit Alexa-488-conjugated secondary antibody (1:1000, Cell Signaling Technology, Danvers, MA) at 4˚C in the dark. After three washes, the cells were resuspended in 200 µl of PBS containing 1% BSA for detection in a NovoCyte flow cytometer (ACEA Biosciences, San Diego, CA) utilizing laser excitation and emission wavelengths of 488 nm and 519 nm, respectively. For each assay point, approximately 15,000 cellular events were collected, and the total fluorescence intensity of positive expression cell population was calculated.

## Plasmid constructs of 5-HT$_7$ receptor

A plasmid encoding the 5-HT$_7$ receptor was obtained from PRESTO-Tango Kit produced by Addgene (Watertown, MA). 5-HT$_7$ coding sequence was amplified and ligated into the pRluc8-N1 vector to produce WT 5-HT$_7$-Rluc8. Mutant 5-HT$_7$-Rluc8 receptors were generated from this plasmid using the Quikchange mutagenesis kit (Agilent Technologies, Santa Clara, CA). A plasmid encoding the Nluc-EPAC-VV cAMP sensor was kindly provided by Kirill Martemyanov (The Scripps Research Institute, Jupiter, FL) and has been described previously. (*Masuho et al., 2015*) All plasmid constructs were verified by DNA sequencing.

## Cell transfection

HEK293 cells cultured in 6-well plates were transiently transfected with the above plasmids (3.0 µg DNA) in growth medium using linear polyethyleneimine MAX (Polysciences, Warrington, PA) at an N/P ratio of 20 and used for experimentation 12–48 hr thereafter.

## BRET cAMP and trafficking assays

HEK 293 cells transiently transfected with WT or mutant 5-HT$_7$ receptor plasmids plus either the Nluc-EPAC-VV cAMP sensor (at a 15:1 ratio) or Venus-kras (at a 1:8 ratio) (*Tian et al., 2017*) were incubated for 24 hr. After washing twice with PBS, they were transferred to opaque black 96-well plates. Steady-state BRET measurements were made using a Mithras LB940 photon-counting plate reader (Berthold Technologies GmbH, Bad Wildbad, Germany). Furimazine (NanoGlo; 1:1000, Promega) for cAMP measurement or coelenterazine h (5 µM; Nanolight, Pinetop, AZ) for trafficking assay was added followed by BRET signal detection and calculation at an emission intensity of 520–545 nm divided by that of 475–495 nm.

## Data and materials availability

The open source code is available at GitHub (*Zhou, 2019*; copy archived at https://github.com/elifesciences-publications/RRCS). For availability of codes that were developed in-house, please contacts the corresponding authors. All data are available in the main text or the source data.

## Acknowledgements

This work was partially supported by National Natural Science Foundation of China grants 31971178 (SZ), 81872915 (M-WW), 21704064 (QZ), 81573479 (DHY) and 81773792 (DHY), National Key R and D Program of China grants 2016YFC0905900 (SZ) and 2018YFA0507000 (SZ and M-WW), National Mega R and D Program for Drug Discovery grants 2018ZX09711002–002–005 (DHY) and 2018ZX09735–001 (M-WW), Shanghai Science and Technology Development Fund grant 16ZR1407100 (ATD), Novo Nordisk-CAS Research Fund grant NNCAS-2017–1-CC (DHY), the Medical Research Council MC_U105185859 (MMB), National Institutes of Health Grants GM130142 (NAL) and annual overhead support from ShanghaiTech University and Chinese Academy of Sciences. We thank A Sali, MA Hanson and AJ Venkatakrishnan for valuable discussions, YM Xu for technical assistance, and AP IJzerman for providing the WT A$_{2A}$R plasmid.

## Additional information

### Funding

| Funder | Grant reference number | Author |
| --- | --- | --- |
| Medical Research Council | MC_U105185859 | M Madan Babu |
| Novo Nordisk-CAS Research | NNCAS-2017-1-CC | Dehua Yang |
| Young Talent Program of Shanghai | | Suwen Zhao |
| Shanghai Science and Technology Development Fund | 16ZR1448500 | Suwen Zhao |
| Shanghai Science and Technology Development Fund | 16ZR1407100 | Antao Dai |
| National Natural Science Foundation of China | 21704064 | Qingtong Zhou |
| National Natural Science Foundation of China | 81573479 | Dehua Yang |
| National Natural Science Foundation of China | 81773792 | Dehua Yang |
| National Natural Science Foundation of China | 31971178 | Suwen Zhao |
| National Mega R&D Program for Drug Discovery | 2018ZX09735-001 | Ming-Wei Wang |
| National Key R&D Program of China | 2016YFC0905900 | Suwen Zhao |

| | | |
|---|---|---|
| National Mega R&D Program for Drug Discovery | 2018ZX09711002-002-005 | Dehua Yang |
| National Key R&D Program of China | 2018YFA0507000 | Ming-Wei Wang Suwen Zhao |
| National Natural Science Foundation of China | 81872915 | Ming-Wei Wang |
| National Institute of General Medical Sciences | GM130142 | Nevin A Lambert |

The funders had no role in study design, data collection and interpretation, or the decision to submit the work for publication.

## Author contributions

Qingtong Zhou, Conceptualization, Resources, Data curation, Software, Formal analysis, Funding acquisition, Validation, Investigation, Visualization, Methodology, Writing - original draft, Writing - review and editing; Dehua Yang, Resources, Data curation, Formal analysis, Funding acquisition, Validation, Writing - review and editing; Meng Wu, Resources, Data curation; Yu Guo, Software; Wanjing Guo, Li Zhong, Xiaoqing Cai, Nevin A Lambert, Data curation, Validation; Antao Dai, Data curation, Funding acquisition, Validation; Wonjo Jang, Validation; Eugene I Shakhnovich, Zhi-Jie Liu, Raymond C Stevens, Provided academic support; M Madan Babu, Conceptualization, Funding acquisition, Writing - original draft, Writing - review and editing; Ming-Wei Wang, Supervision, Funding acquisition, Writing - original draft, Writing - review and editing; Suwen Zhao, Conceptualization, Resources, Formal analysis, Supervision, Funding acquisition, Writing - original draft, Project administration, Writing - review and editing

## Author ORCIDs

Qingtong Zhou https://orcid.org/0000-0001-8124-3079
Dehua Yang http://orcid.org/0000-0003-3028-3243
Eugene I Shakhnovich https://orcid.org/0000-0002-4769-2265
Zhi-Jie Liu https://orcid.org/0000-0001-7279-2893
Nevin A Lambert https://orcid.org/0000-0001-7550-0921
Suwen Zhao https://orcid.org/0000-0001-5609-434X

## Decision letter and Author response

Decision letter https://doi.org/10.7554/eLife.50279.sa1
Author response https://doi.org/10.7554/eLife.50279.sa2

# Additional files

## Supplementary files

- Supplementary file 1. Key resource table.
- Transparent reporting form

## Data availability

All data generated or analysed during this study are included in the manuscript and supporting files. Source data files have been provided for Figures 1, 2, 6 and 7.

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
