## [Decision Letter]

**Acceptance summary:**

This manuscript reports the structural analysis of class A GPCR activation mechanism by developing a novel residue-residue contact score (RRCS) and comparing this score between the active and inactive class A GPCR structures. From this analysis, the authors propose an activation pathway from extracellular ligand binding to the intracellular coupling of signal transducers like G proteins. The proposed pathway involves four clusters (layers) of residues. This proposed activation mechanism is supported by the accompanying mutagenesis experiments.

**Decision letter after peer review:**

Thank you for submitting your article "Universal activation mechanism of class A GPCRs" for consideration by *eLife*. Your article has been reviewed by three peer reviewers, one of whom is a member of our Board of Reviewing Editors, and the evaluation has been overseen by Olga Boudker as the Senior Editor. The reviewers have opted to remain anonymous.

The reviewers have discussed the reviews with one another and the Reviewing Editor has drafted this decision to help you prepare a revised submission.

Essential revisions:

1) There are more than 800 different GPCRs in human body, 45 structures were resolved in class A, and only 4 pairs of single static structures have been analyzed in details in this report. While the analysis aiming to reveal a shared GPCR activation mechanism is a meaningful attempt, it is premature to claim a "universal" activation mechanism. It would be surprising if the activation elements involved in Gs coupling and Gi coupling receptors are entirely conserved, given that the dramatic difference in the outward movement in the cytoplasmic end of TM6 and that the coupling of GPCRs with different intracellular transducers is subjected to different conformational changes. At best we can hope that the proposed mechanism provides a framework and a common language for future structural/mechanistic analysis of GPCRs, and different GPCRs (especially GPCRs of the A family) may to variously degrees and in different aspects resemble proposed mechanism. The entire manuscript, especially the title and the Abstract, needs revision in recognition of this understanding.

2) Within this four detailed analyzed structure, RHO actually doesn't have agonist. It is not justified to include RHO as an example for analysis.

3) Can the authors based on their proposal provide explanation of the basis for GPCR coupling to different G proteins using the RRCS analysis?

4) The author always assumed that ligand binding is located above the highly conserved W6.48, but there are more and more new structures indicated that the ligand could actually diffuse as deep as below D2.50. This is represented by the structure of LTB4 receptor (Nat. Chem. Biol. 14: 262-269) as well as several others. The revision should discuss how a deeper ligand binding would affect their conclusion.

5) Many non-orthosteric ligand binding site have been captured in crystal structures (Sci Rep. 2019; 9: 6180; Front Pharmacol. 2018; 9: 128). How does the proposed activation mechanism provides new insight to these non-orthosteric ligand bindings.

6) The activation mechanism and G protein specificity have been extensively discussed previously (e.g. Nature Structural and Molecular Biology 25(2):185-194 (2018); Nature 545: 317-322 (2017); Nature 536: 484-487 (2016); Nature Chemical Biology 14: 1059-1066(2018); Chem. Rev.20171171139-155). A summary of these discussions should be provided in an expanded Introduction, and the proposed mechanism should be contextualized with respect to the current understanding. Related, the citation of 19 references in one stroke is highly unconventional; this should be revised with a more nuanced account of the current structural understanding of GPCRs.

7) Certain aspects of the proposed model have already been reported, and the authors in the revision should make a special effort in making sure that these reports are properly cited.

8) The description for the four layer elements in the activation pathway as "modular" is quite misleading. "Modular" means isolated and functionally independent, like classic DNA-binding domain and transcriptional activation domain in transcription factors, which can function separately and can be exchanged between transcriptional factors. In this paper, the modality of the four layers in GPCR activation has not been demonstrated. It is not clear whether layer elements can be exchanged between different receptors.

9) Compared to the diverse residue types in the Class A GPCR activation pathways, residues involved in class B GPCR activation appeared to be much more conserved as presented in the recent PTH1R receptor. Discussing the difference and similarity of class A and class B GPCR activation will broaden the interests and impact of the paper.

10) The authors introduced biological mutation experiment to support their ideas. It should be made clear that a positive result can only show the involvement of the mutated residue in GPCR activation, but cannot confirm a residue-residue interaction. With this in mind, the claims associated with the mutation data should be re-calibrated.

11) The paper will be much strengthened if the mutagenesis experiments performed on A2aR are also performed for one of representative Gi coupled receptors. (Probably not the whole set of mutants but the representative CAM and CIM for a Gi coupled receptor.)

12) To confirm the predicted residue-residue interactions, it would be highly desirable to use NMR to show correlated signal changes of the residue pairs.

---

## [Author Response]

Essential revisions:1) There are more than 800 different GPCRs in human body, 45 structures were resolved in class A, and only 4 pairs of single static structures have been analyzed in details in this report. While the analysis aiming to reveal a shared GPCR activation mechanism is a meaningful attempt, it is premature to claim a "universal" activation mechanism. It would be surprising if the activation elements involved in Gs coupling and Gi coupling receptors are entirely conserved, given that the dramatic difference in the outward movement in the cytoplasmic end of TM6 and that the coupling of GPCRs with different intracellular transducers is subjected to different conformational changes. At best we can hope that the proposed mechanism provides a framework and a common language for future structural/mechanistic analysis of GPCRs, and different GPCRs (especially GPCRs of the A family) may to variously degrees and in different aspects resemble proposed mechanism. The entire manuscript, especially the title and the Abstract, needs revision in recognition of this understanding.

We thank the reviewers for the comments. In the revised manuscript, we have reworked on the title, the Abstract and related contents to ensure the accuracy and clarity.

We discuss possible misunderstandings of our work below:

First, we analyzed 6 inactive *vs*. fully active pairs of structures in detail (listed in Author response table 1, inactive state is in orange) but not 4 pairs, as shown in Figure 2D, Figure 2—figure supplement 1, and Results paragraph two. These structures were picked based on their high quality. That is, they are all of high-resolution, no mutation or only one mutation in the transmembrane domain which is known not to affect receptor signaling. These 6 pairs were also selected for analysis by the previous research [Venkatakrishnan, et al. Nature 536.7617 (2016): 484.].

**Author response table 1. resptable1:** List of twelve structures of six receptors.

GPCR	PDB code	Resolution (Å)	Ligand G protein/G protein memetic	State	Mutation in the TM region
bRho	1GZM	2.7	11-cis-retinal (inverse agonist, covalently bound)	Inactive	No mutation
3PQR	2.9	All-trans-retinal (agonist, covalently bound) GαCT2 (G protein peptide)	Active	No mutation
β_2_AR	2RH1	2.4	Carazolol (inverse agonist)	Inactive	No mutation
3SN6	3.2	BI-167107 (agonist) G_s_ (G protein)	Active	No mutation
M2R	3UON	3.0	3-quinuclidinyl-benzilate (antagonist)	Inactive	No mutation
4MQS	3.5	Iperoxo (agonist) Nb9-8 (nanobody)	Active	No mutation
μOR	4DKL	2.8	β-funaltrexamine (antagonist)	Inactive	No mutation
5C1M	2.1	BU72 (agonist) Nb39 (nanobody)	Active	No mutation
A_2A_R	3EML	2.6	ZM241385 (antagonist)	Inactive	No mutation
5G53	3.4	NECA (agonist) Mini-G_s_ (engineered G protein)	Active	No mutation
κOR	4DJH	2.9	JDTic (antagonist)	Inactive	I135L (3×29)
6B73	3.1	MP1104 (agonist) Nb39 (nanobody)	Active	I135L (3×29)

More importantly, beyond the analysis on the six receptors, we performed a class-wide RRCS comparison between the 142 inactive and 27 active state class A GPCR structures to identify residue pairs showing statistically significant different RRCS scores (P<0.001; two sample *t*-test) upon activation (Figure 2D, Figure 2—source data 2, Materials and methods). The combination of such an analysis and the across-class comparison could take the full advantage of tremendous successes of GPCR structural biology in the past decade, and only this exercise would give us the confidence to claim what we discovered are common to diverse class A GPCRs.

Second, the common activation pathway we discovered is a shared portion among G_s_, G_i_, and arrestin activation pathways of various receptors stimulated by various agonists. At the same time, receptors that do have their own receptor-, agonist-, and effector-specific pathways are not the focus of this study. Therefore, we totally agree that activation elements involved in G_s_ and G_i_ coupling are NOT entirely conserved (*i.e.*, biased signaling), and we had avoided to use this term in the manuscript.

2) Within this four detailed analyzed structure, RHO actually doesn't have agonist. It is not justified to include RHO as an example for analysis.

We agree with the reviewers on the importance of structure selection to study receptor activation.

As listed in the Author response table 1, we used a high quality (2.9 Å resolution) structure (PDB: 3PQR) as the fully active state structure of Rho, which has all-trans-retinal (i.e., agonist) covalently bound.

In fact, we did investigate a low resolution (4.5 Å for the overall complex, in local resolution map, the receptor part has even a lower resolution) cryo-EM structure of the Rho-G_i_ complex (PDB: 6CMO), in which agonist is missing. It serves as a negative control for our analysis. We noticed that 6CMO has 8 pairs of residues that do not match the common activation pathway (*i.e.*, 8 outliers out of 34 residues pairs), and we attributed these mismatches to the rather low resolution of the structure and probably the missing of agonist. The positive control is a 3.1 Å crystal structure of the Rho-mini-G_o_ complex (PDB: 6FUF), for which no outliers were observed (Figure 4, Figure 4—figure supplement 1 and Figure 4—figure supplement 2). These results reflect that our RRCS approach is dependent on the high-quality structures.

3) Can the authors based on their proposal provide explanation of the basis for GPCR coupling to different G proteins using the RRCS analysis?

This is a good question. After we developed the RRCS analysis, we have been trying to find possible applications for this method. In theory, this method could be used to analyze any system that have big or subtle but significant conformational changes. The limit is that it really relies on the accuracy of side chain conformations, thus high-resolution structures are the key.

We did apply the RRCS analysis to study G protein selectivity and failed to reach any meaningful conclusions. This may be caused by following reasons:

a) Lack of high resolution GPCR-G protein complex structures. Author response table 2 lists all available GPCR-G protein complexes to date (14 structures from 10 receptors). However, 6 of them have poor resolution (lines in gray) thus are not qualified for the RRCS analysis. For the other cryo-EM structures, carefully checking the local resolution for the GPCR-G protein interface but not directly use the overall resolution of the whole complex is need before doing any RRCS calculation, since in general, large G_β_ protein has the highest resolution in the complex. Hopefully, with more high-resolution complexes determined, RRCS analysis may help answer the question of G protein selectivity.

**Author response table 2. resptable2:** List of GPCR-G protein complex structures.

Receptor	PDB code	Method	Resolution (Å)	Ligand + G protein/G protein memetic
A_2A_R	5G53	X-ray	3.4	NECA + Mini-G_s_
A_2A_R	6GDG	cryo-EM	4.1	NECA + Mini-G_s_
β_2_AR	3SN6	X-ray	3.2	BI-167107 + G_s_
β_2_AR	6E67	X-ray	3.7	BI-167107 + Fused G_s_ C-terminal
Rho	6CMO	cryo-EM	4.5	G_i_
Rho	6FUF	X-ray	3.1	all-trans-retinal +Mini-G_o_
μOR	6DDE/6DDF	cryo-EM	3.5	DAMGO + G_i_
A_1A_R	6D9H	cryo-EM	3.6	Adenosine + G_i2_
5-HT_1B_	6G79	cryo-EM	3.8	Donitriptan + G_o_
CB1	6N4B	cryo-EM	3.0	MDMB-Fubinaca + G_i_
NTSR1	6OSA	cryo-EM	3.0	JMV449 + G_i1_ (NC state)
NTSR1	6OS9	cryo-EM	3.0	JMV449 + G_i1_ (C state)
M2R	6OIK	cryo-EM	3.6	Iperoxo + LY2119620 +G_oA_
M1R	6OIJ	cryo-EM	3.3	Iperoxo + G_11_

b) It is known that G-protein activation is a complex process^1, 2^ which may have several states. Taking NTSR1–G_i1_ complexes for example, a canonical-state and a non-canonical state of G protein were determined showing different hNTSR1–G_i1_ interfaces. The complexity of this dynamic process also increases the difficulty of RRCS analysis.

c) The G protein coupling region is much more diverse in sequence compared to intracellular half of the TM domain in which our common activation pathway locates. Thus, different receptors may recognize different positions of the G-protein through distinct residues, like multiple keys (receptors) opening the same lock (G protein) using non-identical cuts, as a previous study pointed out.^3^

4) The author always assumed that ligand binding is located above the highly conserved W6.48, but there are more and more new structures indicated that the ligand could actually diffuse as deep as below D2.50. This is represented by the structure of LTB4 receptor (Nat. Chem. Biol. 14: 262-269) as well as several others. The revision should discuss how a deeper ligand binding would affect their conclusion.

We thank the reviewers for providing the valuable insights. We took this suggestion and added analysis for several crystal structures (PDB: 5X33, 6K1Q and 6BQH, Author response table 3), the ligand of which diffuses as deep as below D^2.50^. These ligands are all inverse agonists that touch/occupy the Na^+^ pocket. By taking the space of the Na^+^, they prevented the collapse of Na^+^ pocket, as well as blocking the rotation of W^6.48^, leading to lock the receptors to an inactive state.

**Author response table 3. resptable3:** List of three inverse agonist-bound structures.

GPCR	PDB code	Ligand	Ligand type	Na^+^ mimic group in ligand	Occupying Na^+^ pocket
BLT1	5X33	BIIL260	Inverse agonist	Yes	Yes
ET_B_	6K1Q	IRL2500	Inverse agonist	No	Yes
5-HT_2C_	6BQH	Ritanserin	Inverse agonist	No	No

These observations are consistent with our activation model and functional study of A_2A_R mutants. As shown in Figure 6, Figure 6—source data 1, Figure 6—source data 2 and Figure 6—figure supplement 1, three mutations of A_2A_R (L48^2x46^R, N280^7x45^R and N284^7x49^K) that form slat bridges with D^2.50^ and stabilize the inactive state were shown to abolish cAMP accumulation as expected.

Interestingly, for D^2.50^, only loss-of-function disease mutations (Figure 7B) or constitutively inactivating mutations (Figure 7C) were observed for 24 receptors, implying that D^2.50^ is indispensable for receptor activation. These finding are also supported by the NMR study of A_2A_R^4^, which demonstrated the role of D52^2.50^ as an allosteric link between the orthosteric drug binding site and the intracellular signaling surface^4^.

5) Many non-orthosteric ligand binding site have been captured in crystal structures (Sci Rep. 2019; 9: 6180; Front Pharmacol. 2018; 9: 128). How does the proposed activation mechanism provides new insight to these non-orthosteric ligand bindings.

We thank the reviewers for raising this point. Exactly as the recommended literature revealed, allosteric modulation of GPCR provides a promising potential of pharmacological intervention, due to the diversity in location, mechanism, and selectivity of allosteric ligands.

A list of reported allosteric modulators among determined structures of class A GPCRs are listed in Author response table 4, with PAMs labeled in green.

**Author response table 4. resptable4:** List of allosteric modulators among determined structures of class A GPCRs.

Location of binding site	Receptor	Ligand	Ligand type	PDB code
Intracellular pocket	β_2_AR	CMPD-15PA	Negative allosteric modulator	5X7D
CCR9	Vercirnon	Antagonist	5LWE
CCR2	CCR2-RA-[R]	Antagonist	5T1A
Extra-helical binding site	C5aR	NDT9513727	Inverse agonist	5O9H
C5aR	avacopan	Allosteric antagonist	6C1R
C5aR	NDT9513727	Allosteric antagonist	6C1Q
β_2_AR	compound-6FA	Positive allosteric modulator	6N48
P2Y1	BPTU	antagonist	4XNV
GPR40	AP8	Full allosteric agonists (agoPAM)	5TZY
GPR40	Compound 1	Full agonist	5KW2
PAR2	AZ3451	Antagonist	5NDZ
Non-canonical binding pocket	GPR40	TAK-875	Ago-allosteric modulator	4PHU
GPR40	MK-8666	Ago-allosteric modulator	5TZR
PAR2	AZ8838	antagonist	5NDD
Extracellular vestibule above the orthosteric site	M2	LY2119620	Positive allosteric modulator	4MQT

As we can see, the locations of binding site are quite diverse for GPCR allosteric modulators, and so do the mechanisms as one can expect.

Here is our brief analysis of the possible mechanisms. A) Directly altering the common activation pathway. For example, for allosteric modulators bound to the intracellular pocket of their receptor (PDB: 5X7D, 5LWE and 5T1A), they exert antagonism by preventing G protein coupling. B) Having receptor-, ligand-, and effector-specific pathways. An example is compound-6FA (in PDB 6N58), which binds to the extrahelical surface of β_2_AR, but also directly interacts with G protein. C) A mixture of the above two mechanisms.

6) The activation mechanism and G protein specificity have been extensively discussed previously (e.g. Nature Structural and Molecular Biology 25(2):185-194 (2018); Nature 545: 317-322 (2017); Nature 536: 484-487 (2016); Nature Chemical Biology 14: 1059-1066(2018); Chem. Rev.20171171139-155). A summary of these discussions should be provided in an expanded Introduction, and the proposed mechanism should be contextualized with respect to the current understanding. Related, the citation of 19 references in one stroke is highly unconventional; this should be revised with a more nuanced account of the current structural understanding of GPCRs.

We thank the reviewer for the comments. We significantly rewrote the Introduction part to reflect the previous understanding of GPCR activation. Many more references are cited now.

7) Certain aspects of the proposed model have already been reported, and the authors in the revision should make a special effort in making sure that these reports are properly cited.

We thank the reviewer for this comment. We have greatly increased the citation and number of references to reflect previous findings on GPCR activation mechanism.

8) The description for the four layer elements in the activation pathway as "modular" is quite misleading. "Modular" means isolated and functionally independent, like classic DNA-binding domain and transcriptional activation domain in transcription factors, which can function separately and can be exchanged between transcriptional factors. In this paper, the modality of the four layers in GPCR activation has not been demonstrated. It is not clear whether layer elements can be exchanged between different receptors.

We changed the word modular to modular nature, we meant each layer has different function and can be activated/inactivated in a rather independent manner, as shown in our mutagenesis study (Figure 6B).

Analyzing residue pairs in each layer of the common activation pathway for intermediate structures (*i.e.*, only agonist but not G protein bound structures) also supports the modular nature of the common activation pathway. As shown in Author response image 1, β_2_AR (3PDS) only has layer 4 in the active state; 5-HT_2B_ (6DS0) has layers 2 and 3 partially activated, while leaving layers 1 and 4 not activated; A_2A_R (2YDV) and LPA_6_ (5XSZ) have layers 1-3 all partially activated, while leaving layer 4 untouched; and US28 (4XT1) has layers 2-4 in fully active state, while layer 1 is largely inactivated.

**Author response image 1. respfig1:** Receptor activation process via intermediate structures (i.e., only agonist but not G protein bound structures) are in receptor- and ligand-specific manner, thereby highlighting the modular nature and complexity of the activation pathways.

9) Compared to the diverse residue types in the Class A GPCR activation pathways, residues involved in class B GPCR activation appeared to be much more conserved as presented in the recent PTH1R receptor. Discussing the difference and similarity of class A and class B GPCR activation will broaden the interests and impact of the paper.

We thank the reviewers for this suggestion. Actually, the idea of applying RRCS on other allosteric systems including class B GPCRs is on our to-do-list. However, we have not done so due to the following reasons:

a) Different from class A GPCRs, only one receptor (GLP-1R) has been determined with both inactive and fully active state structures, while other receptors only have either antagonist-bound inactive structures or G protein-bound active structures.

**Author response table 5. resptable5:** List of available class B GPCRs structures.

Receptor	PDB code	Method	Resolution (Å)	Ligand + G protein/G protein memetic	States
CRF1R	4K5Y	X-ray	2.9	Antagonist CP-376395	Inactive
CRF1R	4Z9G	X-ray	3.2	Antagonist CP-376395	Inactive
CTR	6NIY	cryo-EM	3.3	Peptide ligand + G_s_	Active
CTR	5UZ7	cryo-EM	4.1	Peptide ligand + G_s_	Active
CGRP	6E3Y	cryo-EM	3.3	CGRP + G_s_	Active
GCGR	4L6R	X-ray	3.3	No ligand was seen	Inactive
GCGR	5EE7	X-ray	2.5	Antagonist MK-0893	Inactive
GCGR	5YQZ	X-ray	3.0	Partial agonist NNC1702	Intermediate
GCGR	5XEZ	X-ray	3.0	Negative allosteric modulator NNC0640	Inactive
GCGR	5XF1	X-ray	3.2	Negative allosteric modulator NNC0640	Inactive
GLP-1R	5VAI	cryo-EM	4.1	human GLP-1 + G_s_	Active
GLP-1R	6B3J	cryo-EM	3.3	Exendin-P5 + G_s_	Active
GLP-1R	5NX2	X-ray	3.7	Truncated peptide agonist	Intermediate
GLP-1R	5VEW	X-ray	2.7	PF-06372222	Inactive
GLP-1R	5VEX	X-ray	3.0	NNC0640	Inactive
PTH1R	6FJ3	X-ray	2.5	Peptide agonist ePTH	Intermediate
PTH1R	6NBH	cryo-EM	3.5	A long-acting PTH analog + G_s_	Active
PTH1R	6NBI	cryo-EM	4.0	A long-acting PTH analog + G_s_	Active
PTH1R	6NBF	cryo-EM	3.0	A long-acting PTH analog + G_s_	Active

b) The active state structures of class B GPCRs are all EM structures, their resolutions are generally low compared to that of crystal structures. As mentioned before, the RRCS analysis highly relies on the quality of side chain conformations. More class B structures with resolution higher than 3.0 Å are needed.

c) The primary results of RRCS on class B GPCRs revealed that RRCS could describe the well-known conformational changes including the HETX motif (Nature. 2017;546(7657):248-253^5^) very well (Author response image 2).

**Author response image 2. respfig2:** Calculated RRCS for the HETX motif in class B GPCRs.

The comparison of the activation mechanism between classes A and B GPCRs have been frequently discussed^6^. The most notable conformational change in class B GPCRs is the sharp kink and unwinding in the middle of TM6, as observed for GLP-1R^5^/CTR^7^/CGRP^8^/PTH1R^9^. Another notable difference is the signal initiation and transduction pathway: class A has PIF/CWxP/Na^+^ pocket/DRY/NPxxY motifs, while class B has totally different conserved motifs. Despite these differences, they both couple G protein in similar orientation, highlighting the converge of signal transduction in the cytoplasmic region.

10) The authors introduced biological mutation experiment to support their ideas. It should be made clear that a positive result can only show the involvement of the mutated residue in GPCR activation, but cannot confirm a residue-residue interaction. With this in mind, the claims associated with the mutation data should be re-calibrated.

Thank the reviewer for raising this point. We were very careful when making mechanistic interpretations of the successfully predicted mutants, we always used the words “likely”, “would”, and “probably” to avoid any confirmation of residue-residue interactions, unless such interactions were reported by NMR studies.

11) The paper will be much strengthened if the mutagenesis experiments performed on A2aR are also performed for one of representative Gi coupled receptors. (Probably not the whole set of mutants but the representative CAM and CIM for a Gi coupled receptor.)

Thank the reviewer for the suggestion. We expanded our mutagenesis studies to two additional receptors: G_s_-coupled 5-HT_7_ and G_i_-coupled 5-HT_1B_. The three mutations on the G_s_-coupled 5-HT_7_ were designed and validated by Dr. Nevin Lambert (who designed the mutations by directly following our designs on A_2A_R reported in the preprint of this manuscript. We did not interact with each other before this paper was submitted to the preprint server). The four mutations on the G_i_-coupled 5-HT_1B_ were validated by ourselves. All the mutations are remarkably consistent with what we discovered in A_2A_R. We added a new figure (Figure 6—figure supplement 2) to show the new mutagenesis data.

12) To confirm the predicted residue-residue interactions, it would be highly desirable to use NMR to show correlated signal changes of the residue pairs.

This is an excellent suggestion. Exactly, NMR study could provide comprehensive and precise information on residue contacts changes upon receptor activation. Although we were unable to perform NMR study due to 2-month time limit and lack of expertise, we collected and analyzed published NMR data, which covered several regions in the common activation pathway, including residues at 5.57, 3.40, 5.58, 7.53, the Na^+^ pocket, TM6 and DRY motif. The following is a brief summary of our analysis.

Isogai et al. in the NMR study of turkey β_1_-adrenergic receptor (β_1_AR)^10^ demonstrated that chemical shifts of the labeled V226^5.57^ correlate linearly with ligand efficacies of the G protein pathway, thereby proving the involvement of residue 5.57 in receptor activation. The same NMR study also showed the participation of three residues (V129^3.40^, A227^5.58^, and L343^7.53^) in receptor activation through NMR response to ligand binding and point mutations (V129^3.40^I, A227^5.58^Y, and L343^7.53^Y) These results provide experimental evidence at high resolution of an extensive signal transduction network that connects the ligand binding site to the intracellular sides of TM5, TM6, and TM7.

Notably, 5.57, 3.40, 5.58 and 7.53 are nodes in our common activation pathway. Exactly as shown in Figure 5 and Figure 5—figure supplement 1, receptor activation involves the elimination of TM3-TM6 contacts, formation of TM3-TM7 and TM5-TM6 contacts, reflecting the outward movement of the cytoplasmic end of TM6 away from TM3, the inward movement of TM7 towards TM3 and the repacking of TM5 and TM6.

By using stable isotope NMR study for A_2A_ adenosine receptor (A_2A_R)^4^,Eddy et al. observed the interplay of the ‘‘toggle switch’’ W246^6.48^ with the allosteric center at D52^2.50^. The observation is modelled in Figures 3 and 5 of our manuscript.

**References**

1. Kato, H. E.; Zhang, Y.; Hu, H.; Suomivuori, C. M.; Kadji, F. M. N.; Aoki, J.; Krishna Kumar, K.; Fonseca, R.; Hilger, D.; Huang, W.; Latorraca, N. R.; Inoue, A.; Dror, R. O.; Kobilka, B. K.; Skiniotis, G., Conformational transitions of a neurotensin receptor 1-Gi1 complex. Nature 2019, 572 (7767), 80-85.

2. Liu, X.; Xu, X.; Hilger, D.; Aschauer, P.; Tiemann, J. K. S.; Du, Y.; Liu, H.; Hirata, K.; Sun, X.; Guixa-Gonzalez, R.; Mathiesen, J. M.; Hildebrand, P. W.; Kobilka, B. K., Structural Insights into the Process of GPCR-G Protein Complex Formation. Cell 2019, 177 (5), 1243-1251 e12.

3. Flock, T.; Hauser, A. S.; Lund, N.; Gloriam, D. E.; Balaji, S.; Babu, M. M., Selectivity determinants of GPCR-G-protein binding. Nature 2017, 545 (7654), 317-322.

4. Eddy, M. T.; Lee, M. Y.; Gao, Z. G.; White, K. L.; Didenko, T.; Horst, R.; Audet, M.; Stanczak, P.; McClary, K. M.; Han, G. W.; Jacobson, K. A.; Stevens, R. C.; Wuthrich, K., Allosteric Coupling of Drug Binding and Intracellular Signaling in the A2A Adenosine Receptor. Cell 2018, 172 (1-2), 68-80 e12.

5. Zhang, Y.; Sun, B.; Feng, D.; Hu, H.; Chu, M.; Qu, Q.; Tarrasch, J. T.; Li, S.; Sun Kobilka, T.; Kobilka, B. K.; Skiniotis, G., Cryo-EM structure of the activated GLP-1 receptor in complex with a G protein. Nature 2017, 546 (7657), 248-253.

6. Jazayeri, A.; Rappas, M.; Brown, A. J. H.; Kean, J.; Errey, J. C.; Robertson, N. J.; Fiez-Vandal, C.; Andrews, S. P.; Congreve, M.; Bortolato, A.; Mason, J. S.; Baig, A. H.; Teobald, I.; Dore, A. S.; Weir, M.; Cooke, R. M.; Marshall, F. H., Crystal structure of the GLP-1 receptor bound to a peptide agonist. Nature 2017, 546 (7657), 254-258.

7. Liang, Y. L.; Khoshouei, M.; Radjainia, M.; Zhang, Y.; Glukhova, A.; Tarrasch, J.; Thal, D. M.; Furness, S. G. B.; Christopoulos, G.; Coudrat, T.; Danev, R.; Baumeister, W.; Miller, L. J.; Christopoulos, A.; Kobilka, B. K.; Wootten, D.; Skiniotis, G.; Sexton, P. M., Phase-plate cryo-EM structure of a class B GPCR-G-protein complex. Nature 2017, 546 (7656), 118-123.

8. Liang, Y. L.; Khoshouei, M.; Deganutti, G.; Glukhova, A.; Koole, C.; Peat, T. S.; Radjainia, M.; Plitzko, J. M.; Baumeister, W.; Miller, L. J.; Hay, D. L.; Christopoulos, A.; Reynolds, C. A.; Wootten, D.; Sexton, P. M., Cryo-EM structure of the active, G(s)- protein complexed, human CGRP receptor. Nature 2018, 561 (7724), 492-+.

9. Zhao, L. H.; Ma, S.; Sutkeviciute, I.; Shen, D. D.; Zhou, X. E.; de Waal, P. W.; Li, C. Y.; Kang, Y.; Clark, L. J.; Jean-Alphonse, F. G.; White, A. D.; Yang, D.; Dai, A.; Cai, X.; Chen, J.; Li, C.; Jiang, Y.; Watanabe, T.; Gardella, T. J.; Melcher, K.; Wang, M. W.; Vilardaga, J. P.; Xu, H. E.; Zhang, Y., Structure and dynamics of the active human parathyroid hormone receptor-1. Science 2019, 364 (6436), 148-153.

10. Isogai, S.; Deupi, X.; Opitz, C.; Heydenreich, F. M.; Tsai, C. J.; Brueckner, F.; Schertler, G. F.; Veprintsev, D. B.; Grzesiek, S., Backbone NMR reveals allosteric signal transduction networks in the beta1-adrenergic receptor. Nature 2016, 530 (7589), 237-41.